# Variability of visual field maps in human early extrastriate cortex challenges the canonical model of organization of V2 and V3

Fernanda Lenita Ribeiro[1,2,3]*, Ashley York[1,2], Elizabeth Zavitz[4,5,6], Steffen Bollmann[3,7], Marcello GP Rosa[4,5†], Alexander Puckett[1,2†]

[1]School of Psychology, The University of Queensland, Brisbane, Australia; [2]Queensland Brain Institute, The University of Queensland, Brisbane, Australia; [3]School of Electrical Engineering and Computer Science, The University of Queensland, Brisbane, Australia; [4]Department of Physiology, Monash University, Melbourne, Australia; [5]Neuroscience Program, Biomedicine Discovery Institute; Monash University, Melbourne, Australia; [6]Department of Electrical and Computer Systems Engineering, Monash University, Clayton, Australia; [7]Queensland Digital Health Centre, The University of Queensland, Brisbane, Australia

*For correspondence:
fernanda.ribeiro@uq.edu.au

†Joint senior authors

**Abstract** Visual field maps in human early extrastriate areas (V2 and V3) are traditionally thought to form mirror-image representations which surround the primary visual cortex (V1). According to this scheme, V2 and V3 form nearly symmetrical halves with respect to the calcarine sulcus, with the dorsal halves representing lower contralateral quadrants, and the ventral halves representing upper contralateral quadrants. This arrangement is considered to be consistent across individuals, and thus predictable with reasonable accuracy using templates. However, data that deviate from this expected pattern have been observed, but mainly treated as artifactual. Here, we systematically investigate individual variability in the visual field maps of human early visual cortex using the 7T Human Connectome Project (HCP) retinotopy dataset. Our results demonstrate substantial and principled inter-individual variability. Visual field representation in the dorsal portions of V2 and V3 was more variable than in their ventral counterparts, including substantial departures from the expected mirror-symmetrical patterns. In addition, left hemisphere retinotopic maps were more variable than those in the right hemisphere. Surprisingly, only one-third of individuals had maps that conformed to the expected pattern in the left hemisphere. Visual field sign analysis further revealed that in many individuals the area conventionally identified as dorsal V3 shows a discontinuity in the mirror-image representation of the retina, associated with a Y-shaped lower vertical representation. Our findings challenge the current view that inter-individual variability in early extrastriate cortex is negligible, and that the dorsal portions of V2 and V3 are roughly mirror images of their ventral counterparts.

## Editor's evaluation

This fundamental work advances our understanding of the retinotopic organization of early visual cortex in humans, and in particular, patterns of variability in the organization of V2 and V3 in normal populations. The evidence supporting the conclusions is compelling, with rigorous retinotopic analysis across a large publicly-available dataset. The work will be of broad interest to vision scientists, as well as those interested in inter-individual variability and neural fingerprinting using fMRI.

## Introduction

Non-invasive imaging has been instrumental in mapping the topographic organization of human visual cortex (*Wandell and Winawer, 2011*). The visual field maps in early visual areas (V1, V2, and V3) have been reported to be remarkably consistent across people, and predictable with reasonable accuracy using a template (*Benson et al., 2014*; *Benson et al., 2012*; *Schira et al., 2010*). While V1 contains a complete, first-order (continuous) representation of the contralateral visual hemifield, areas V2 and V3 form second-order (discontinuous) representations (*Rosa, 2002*). In these areas, a field discontinuity near the horizontal meridian splits the maps into upper and lower field representations that are only connected at the foveal confluence (*Figure 1a, b*). Accordingly, in parcellation schemes (*Glasser et al., 2016*; *Wang et al., 2015*), early visual areas form concentric bands, arranged in nearly symmetrical halves with respect to the calcarine sulcus. These bands, each containing the representation of a contralateral visual field quadrant, are referred to as the dorsal and ventral portions of V2 and V3 (*Figure 1a*). However, observations originating in several laboratories have indicated departures from this pattern, particularly in the dorsal region (*Allen et al., 2022*; *Arcaro and Kastner, 2015*; *Benson and Winawer, 2018*; *Van Essen and Glasser, 2018*). Even so, small-sized datasets, variability in acquisition sites and protocols, and methodological constraints have limited the investigation of this variability. As a result, no consensus exists about deviations from the canonical mirror-symmetrical organization of V2 and V3.

In humans, empirical visual field mapping using functional MRI (fMRI) is the primary means of delineating precise visual area boundaries in individuals. Visual field maps are typically defined in polar coordinates, resulting in two maps: one representing polar angle (or clock position) and the other eccentricity (or distance away from the fixation point) (*Wandell and Winawer, 2011*). In primates, isoangle bands representing the vertical and the horizontal meridians are thought to delineate boundaries between V1 and V2, V2 and V3, and V3 and higher-order visual areas (*Figure 1b*). Particularly, in human probabilistic maps, boundaries between the dorsal portions of early visual areas are roughly mirror images of their ventral counterparts (*Figure 1c*). Nevertheless, boundaries that deviate from the expected ones exist, but these have been mainly treated as artifactual, with researchers often overlooking the irregularities by simply drawing the boundaries to resemble that of a canonical map as best as possible (e.g., *Figure 1d*). Here, it may be important to remark that the border between the dorsal parts of V2 and V3 is well known to be variable in other mammals, and that it typically does not coincide with the representation of the horizontal meridian (see *Rosa and Manger, 2005* for review).

Although previous reports of individual variability in the dorsal portion of human early visual cortex were primarily anecdotal (*Allen et al., 2022*; *Arcaro and Kastner, 2015*; *Benson and Winawer, 2018*; *Van Essen and Glasser, 2018*), a recently developed deep learning model predicts that individual variability in retinotopy exists, and that this is correlated with variations in gross anatomy (e.g., the pattern of sulci and gyri) (*Ribeiro et al., 2021*). Moreover, studies modeling the formation of retinotopic maps in non-human primates also indicate that different variants could develop based on the application of similar developmental rules (*Yu et al., 2020*).

Motivated by these findings, here we systematically investigate individual variability in visual field maps of human early visual cortex using a recently released, large-scale dataset: the 181 participants, 7T Human Connectome Project (HCP) retinotopy dataset (*Benson et al., 2018*). Our aims were to quantify the level of individual variability throughout early visual cortex (V1–V3) and to determine whether there are common modes of retinotopic organization that differ from the established view (i.e., whether individual retinotopic maps differ from a template in similar ways). Our results demonstrate that the dorsal portions of human early visual areas are more heterogeneous than previously acknowledged and challenge the current view that individual differences in retinotopic organization reflect experimental artifacts that may be dismissed for practical purposes.

## Results

### Individual variability in retinotopy

We defined an individual variability metric to quantify how variable visual field maps are across visual areas (V1, V2, and V3), portions (dorsal and ventral), and hemispheres (left and right) in human early visual cortex. First, we computed the average visual field maps across all 181 individuals from the HCP retinotopy dataset for both left and right hemispheres. Then, we iteratively calculated the difference

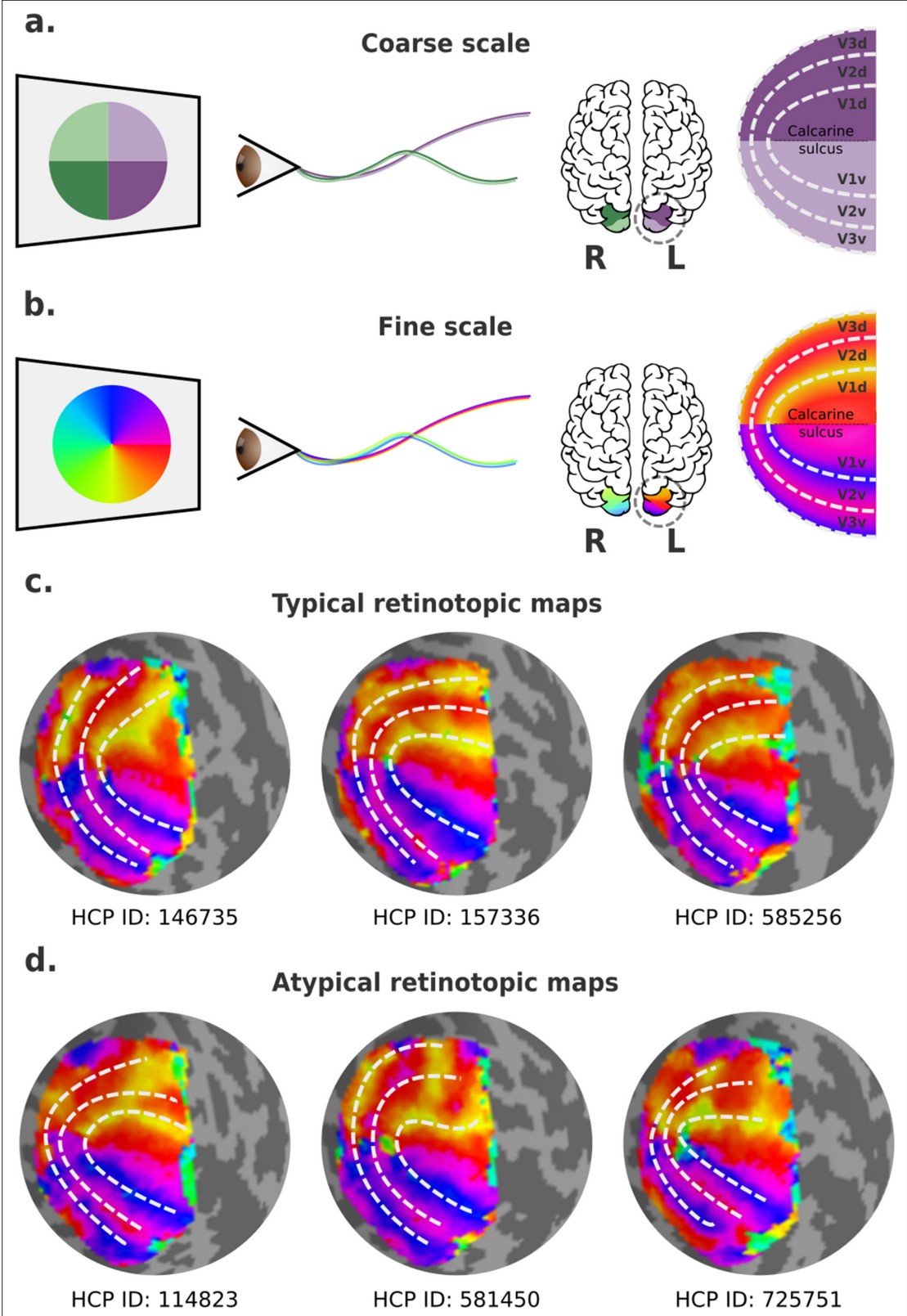

**Figure 1.** Visual field mapping in the human early visual cortex. (**a**) Coarse scale visual field mapping in the early visual cortex. The left (L) hemisphere maps the right visual field, and the right (R) hemisphere maps the left visual field. The dorsal portion of early visual areas maps the lower hemifield, and the ventral portion the upper field. (**b**) Fine scale visual field mapping with visual field maps represented in polar angles (0–360°). In this model, the vertical (90° or 270°) and horizontal meridians (0° for the left and 180° for the right hemispheres) delineate boundaries between visual areas. (**c**) Three

*Figure 1 continued on next page*

*Figure 1 continued*

'classical' polar angle maps, obtained from the left hemispheres of three individuals in the Human Connectome Project (HCP) retinotopy dataset, which conform to the traditional model. (**d**) Three polar angle maps that deviate from this pattern, obtained from left hemispheres of three other individuals in the HCP retinotopy dataset. In the latter, the isopolar bands representing the anterior borders of dorsal V3 (V3d) and dorsal V2 (V2d) do not follow the proposed borders of V2 and V3 (dashed lines).

The online version of this article includes the following figure supplement(s) for figure 1:

**Figure supplement 1.** Average retinotopic maps across all 181 individuals from the Human Connectome Project (HCP) retinotopy dataset for both left (LH) and right (RH) hemispheres.

between an individual's visual field map and the average map. Finally, these differences were averaged over all vertices within the dorsal and ventral portions of early visual areas, resulting in one scalar value per individual per visual area, which is our individual variability metric. Therefore, this metric is a proxy for large-scale deviations in visual field mapping. *Figure 2* shows the distribution of individual variability scores across all participants.

We built a linear mixed-effect (LME) model (*Yu et al., 2022*) to test the fixed effects of hemispheres, visual areas, and portions on large-scale individual variability of polar angle (*Table 1*) and eccentricity (*Table 2*) maps. *Table 1* shows statistically significant main effects of all factors on individual variability of polar angle maps. Specifically, polar angle maps of the left hemisphere show higher individual variability than those found in the right hemisphere (mean difference = 3.35, p < 0.001). The dorsal portions of early visual areas are also more variable than the ventral portions (mean difference = 3.30, p < 0.001). Finally, post hoc comparisons of visual areas indicated that V3 has higher individual variability than V2 (mean difference = 1.60, p < 0.001) and V1 (mean difference = 3.99, p < 0.001); V2 also has higher individual variability than V1 (mean difference = 2.38, p < 0.001). For brevity, we only show the main effects in *Table 1*, although we also found statistically significant interactions. Briefly, each visual area in the left hemisphere has significantly higher individual variability than its analogous area in the right hemisphere. In addition, the dorsal portion of each visual area of the left hemisphere is significantly more variable than its dorsal analog in the right hemisphere and the ventral analog of both the left and right hemispheres (for more, see *Supplementary file 1*). These findings suggest

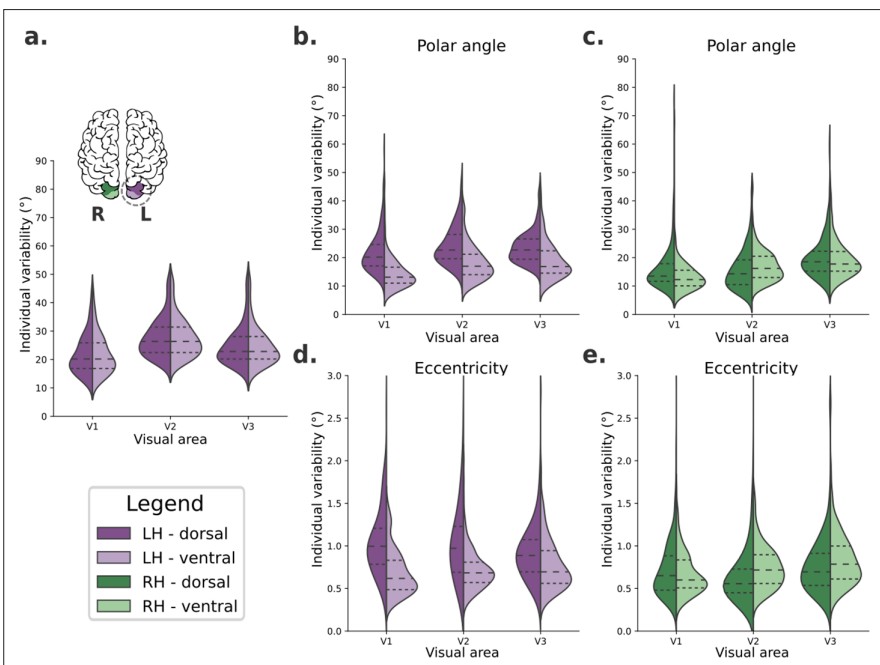

**Figure 2.** Individual variability in visual field maps of early visual areas. (**a**) Hypothetical diagram of symmetrical distributions of individual variability across visual areas. The center and right columns illustrate empirical distributions of individual variability of polar angle (**b, c**) and eccentricity (**d, e**) maps for both dorsal (dark shades) and ventral (lighter shades) portions of early visual areas in left (purple) and right (green) hemispheres.

**Table 1.** Fixed effects parameter estimates for the linear mixed-effect model of individual variability of polar angle maps.

**Polar angle**

| Names | Effect | Estimate | SE | 95% CI Lower | Upper | df | t | p |
|---|---|---|---|---|---|---|---|---|
| Intercept | Intercept | 18.58 | 0.30 | 17.99 | 19.17 | 180 | 61.86 | <0.001 |
| Hemisphere | RH–LH | −3.35 | 0.32 | −3.97 | −2.72 | 181 | −10.47 | <0.001 |
| Visual area (1) | V2–V1 | 2.38 | 0.29 | 1.81 | 2.95 | 210 | 8.21 | <0.001 |
| Visual area (2) | V3–V1 | 3.99 | 0.32 | 3.36 | 4.61 | 187 | 12.54 | <0.001 |
| Portion | Ventral–dorsal | −3.30 | 0.29 | −3.86 | −2.74 | 181 | −11.50 | <0.001 |

SE – standard error; CI – confidence interval.

that individual variability in polar angle representations varies across hemispheres, visual areas, and according to dorsal/ventral locations.

Moreover, *Table 2* shows statistically significant main effects of the hemisphere, visual area, and the visual area portion on individual variability of eccentricity maps. Like polar angle maps, eccentricity maps of the left hemisphere show higher individual variability than those in the right hemisphere (mean difference = 0.14, p < 0.001). The dorsal portion of early visual areas is also more variable than the ventral portion (mean difference = 0.13, p < 0.001). For visual areas, post hoc comparisons indicated that the only statistically significant difference was that of V3 versus V1, with V3 having higher individual variability than V1 (mean difference = 0.05, p < 0.004). In addition, statistically significant interactions were also found (*Supplementary file 2*). Each visual area in the left hemisphere has significantly higher individual variability than analogous areas in the right hemisphere, except for V3. Eccentricity maps of each visual area's dorsal portion in the left hemisphere are significantly more variable than the dorsal counterpart in the right hemisphere, and the ventral analogs in both the left and the right hemispheres.

## Influence of covariates on individual variability of retinotopic maps

Other potential sources of individual variability in retinotopic maps include covariates, such as cortical curvature and mean blood oxygenation level dependent (BOLD) signal. As mentioned in the Materials and methods, these factors are known to be correlated with retinotopy and could explain part of the variability found across visual areas. In *Figure 3*, we show pair-wise correlations among polar angle, eccentricity, curvature, and normalized mean BOLD signal for both V1–3 (main plots) and each visual area separately (inset plots). Across V1–3 (*Figure 3*, main plots), polar angle was significantly correlated with curvature and the mean BOLD signal in both the right and left hemispheres; eccentricity was correlated with the mean BOLD signal in both hemispheres and curvature in the right

**Table 2.** Fixed effects parameter estimates for the linear mixed-effect model of individual variability of eccentricity maps.

**Eccentricity**

| Names | Effect | Estimate | SE | 95% CI Lower | Upper | df | t | p |
|---|---|---|---|---|---|---|---|---|
| Intercept | Intercept | 0.81 | 0.02 | 0.77 | 0.85 | 180 | 41.86 | <0.001 |
| Hemisphere | RH–LH | −0.14 | 0.01 | −0.16 | −0.11 | 181 | −10.91 | <0.001 |
| Visual area (1) | V2–V1 | 0.01 | 0.01 | −0.01 | 0.04 | 402 | 0.98 | 0.326 |
| Visual area (2) | V3–V1 | 0.05 | 0.01 | 0.02 | 0.08 | 182 | 3.24 | 0.001 |
| Portion | Ventral–dorsal | −0.13 | 0.03 | −0.18 | −0.07 | 180 | −4.64 | <0.001 |

SE – standard error; CI – confidence interval.

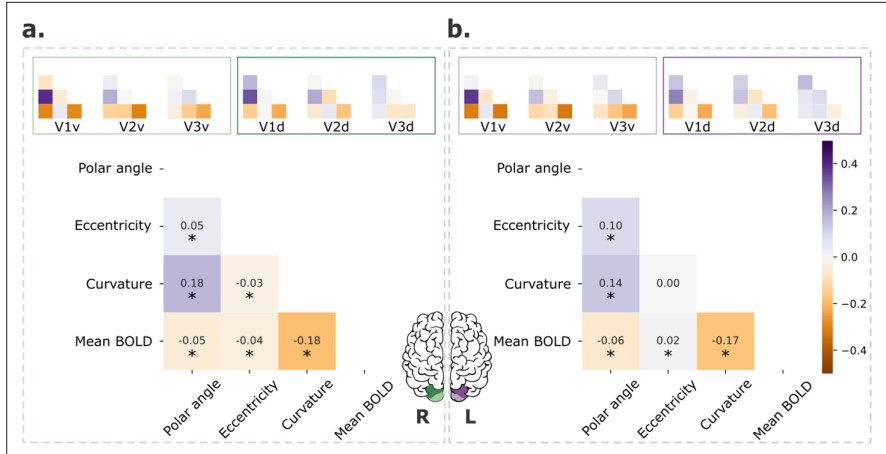

**Figure 3.** Retinotopic maps correlation with covariates. Pair-wise correlations among polar angle, eccentricity, curvature, and normalized mean BOLD signal for early visual areas (V1–3; main plot) and each visual area separately (inset plots), for both left (**a**) and right (**b**) hemispheres. Polar angle maps were converted such that 0° corresponds to the horizontal meridian and 90° corresponds to the upper and lower vertical meridians (*Kurzawski et al., 2022*). Finally, data were concatenated across all participants (*n* = 181). *p < 0.001.

The online version of this article includes the following figure supplement(s) for figure 3:

**Figure supplement 1.** Intra-individual variability in visual field maps of early visual areas.

hemisphere. Moreover, the correlation between these maps varied across visual areas. Specifically, correlations between polar angle and curvature, polar angle and mean BOLD signal, and curvature and mean BOLD signal decreased from V1 to V2 to V3.

Given the meaningful correlation between some of these variables and their varying degree of association across visual areas, we further include individual variability in curvature and mean BOLD signal maps and the intra-individual variability in pRF estimates (a proxy for the reliability of the retinotopic maps; *Figure 3—figure supplement 1*) as covariates in the LME model (*Tables 3 and 4*; *Supplementary files 3 and 4*). Our findings indicate that the main effects found here were not a

**Table 3.** Fixed effects parameter estimates for the linear mixed-effects model of individual variability of polar angle maps using covariates.

The covariates included in the model were intra-individual variability in pRF estimates, and individual variability in curvature and the mean BOLD signal.

**Polar angle**

| Names | Effect | Estimate | SE | 95% CI Lower | Upper | df | t | p |
|---|---|---|---|---|---|---|---|---|
| Intercept | Intercept | 18.58 | 0.27 | 18.06 | 19.11 | 179 | 69.58 | <0.001 |
| Hemisphere | RH–LH | −2.44 | 0.28 | −2.99 | −1.89 | 189 | −8.71 | <0.001 |
| Visual area (1) | V2–V1 | 1.93 | 0.33 | 1.29 | 2.57 | 323 | 5.90 | <0.001 |
| Visual area (2) | V3–V1 | 3.54 | 0.32 | 2.92 | 4.16 | 288 | 11.22 | <0.001 |
| Portion | Ventral–dorsal | −2.12 | 0.29 | −2.69 | −1.54 | 207 | −7.20 | <0.001 |
| Intra-individual variability | Intra-individual variability | 2.90 | 0.16 | 2.58 | 3.22 | 1364 | 17.81 | <0.001 |
| Individual variability in curvature | Individual variability in curvature | 0.40 | 0.14 | 0.12 | 0.68 | 1812 | 2.82 | 0.005 |
| Individual variability in mean BOLD signal | Individual variability in mean BOLD signal | 0.21 | 0.19 | −0.16 | 0.57 | 597 | 1.11 | 0.268 |

SE – standard error; CI – confidence interval.

**Table 4.** Fixed effects parameter estimates for the linear mixed-effects model of individual variability of eccentricity maps using covariates.

The covariates included in the model were intra-individual variability in pRF estimates, and individual variability in curvature and the mean BOLD signal.

**Polar angle**

| Names | Effect | Estimate | SE | 95% CI Lower | Upper | df | t | p |
|---|---|---|---|---|---|---|---|---|
| Intercept | Intercept | 0.81 | 0.01 | 0.78 | 0.84 | 179 | 54.64 | <0.001 |
| Hemisphere | RH–LH | −0.07 | 0.01 | −0.09 | −0.05 | 201 | −6.40 | <0.001 |
| Visual area (1) | V2–V1 | −0.02 | 0.01 | −0.05 | 0.01 | 506 | −1.52 | 0.128 |
| Visual area (2) | V3–V1 | −0.02 | 0.02 | −0.05 | 0.01 | 262 | −1.20 | 0.231 |
| Portion | Ventral–dorsal | −0.07 | 0.02 | −0.11 | −0.03 | 186 | −3.24 | 0.001 |
| Intra-individual variability | Intra-individual variability | 0.21 | 0.01 | 0.20 | 0.23 | 1982 | 26.51 | <0.001 |
| Individual variability in curvature | Individual variability in curvature | 0.01 | 0.01 | −0.00 | 0.02 | 1829 | 1.52 | 0.128 |
| Individual variability in mean BOLD signal | Individual variability in mean BOLD signal | 0.02 | 0.01 | 0.00 | 0.03 | 489 | 2.34 | 0.020 |

SE – standard error; CI – confidence interval.

mere reflection of variation in the reliability of the individual maps and other covariates, at least at large scale. For example, we found that the main effects of all factors (hemispheres, visual areas, and portions) on individual variability of polar angle maps persisted, but the estimated effects were slightly reduced (*Table 3*). To determine whether other properties would predict individual variability in polar angle maps, we included age, gender (as cofactor), and gray matter volume as additional covariates, but did not find significant effects of any of these variables (not shown here, but available as *Supplementary file 5*). For eccentricity, we found the main effects of hemisphere and portion but not visual area after accounting for these covariates (*Table 4*). Crucially, these findings suggest that individual variability in polar angle representations varies as a function of hemispheres, visual areas, and dorsal/ventral locations. These effects are not only due to trivial intra-individual variability in pRF estimates or individual variability in curvature and the mean BOLD signal.

## Common modes of variability

Next, we performed an exploratory analysis to determine whether polar angle maps differ from the average map in similar ways, particularly in the dorsal portion of early visual cortex of the left hemisphere (*Figure 4*). Note that an analogous analysis was performed for eccentricity maps, but in agreement with our statistical models we did not find meaningful differences across eccentricity map clusters (*Figure 4—figure supplement 1*). We computed the extent of overlap between discrete polar angle maps from all possible pairs of individuals using the Jaccard index, resulting in a similarity matrix (*Figure 4a*). Next, we applied a spectral clustering algorithm with a fixed number of clusters equal to 6 (*Figure 4b*). Finally, we averaged the continuous polar angle maps across individuals within each cluster to visualize common patterns of retinotopic organization in the dorsal portion of early visual cortex (*Figure 4c*; see *Figure 4—figure supplement 2* for the complete set of average maps based on *Figure 4c* clustering assignment).

Our findings clearly indicate shared patterns of retinotopic organization that deviate from the canonical polar angle representation in the dorsal portion of early visual cortex (*Figure 1c*). Specifically, average maps from clusters 1 and 5 capture nearly a third of individuals and show canonical polar angle representations, with clear boundaries between V1/V2 and V2/V3 (*Figure 1c* and *Figure 4c*). However, clusters 2, 3, and 4 capture nearly two-thirds of individuals and deviate from this canonical polar angle representation (*Figure 4c*). The average map from cluster 2 shows that the boundaries between V1 and V2, and the most anterior portion of V3 and higher-order visual areas, merge to form a Y-shaped (or forked) lower vertical representation. Clusters 3 and 4 show a truncated V3 boundary,

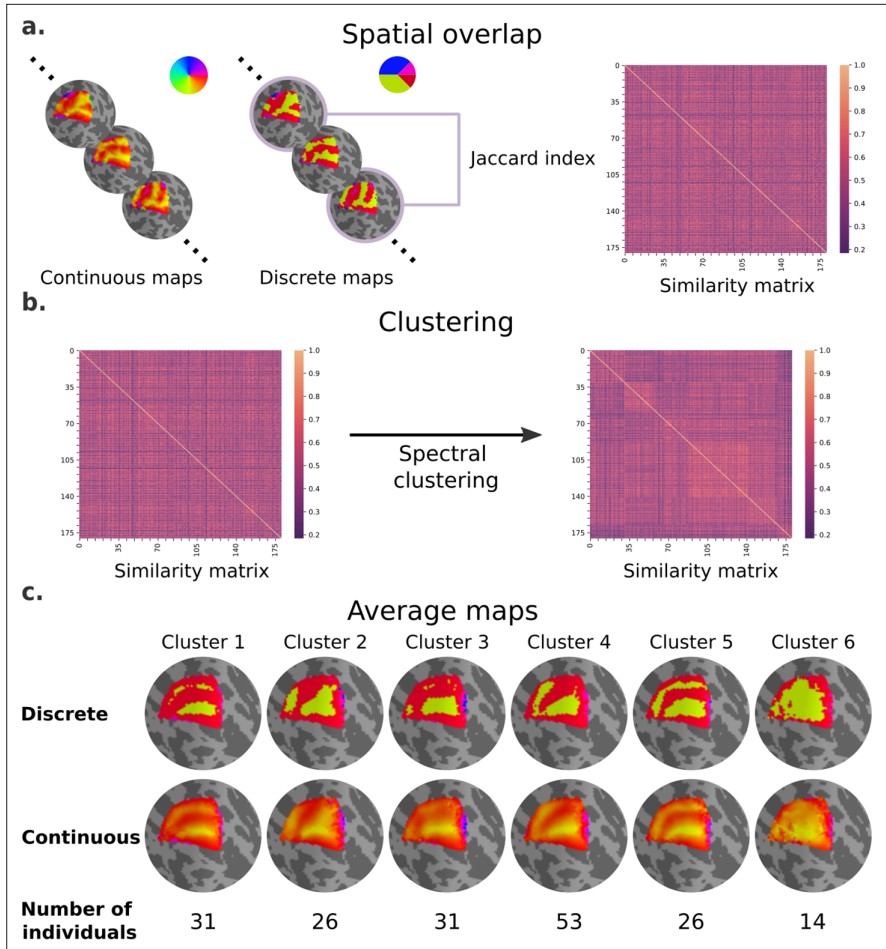

**Figure 4.** Clusters of retinotopic organization in the dorsal portion of early visual cortex. (**a**) Continuous polar angle maps were converted into discrete maps, such that each vertex would be categorized into one out of four possible labels. Spatial overlap between discrete maps was estimated using the Jaccard similarity coefficient from all possible pairs of individuals, resulting in a 181 × 181 similarity matrix. (**b**) Then, we applied a spectral clustering algorithm – setting the number of clusters to 6. (**c**) An average map (discrete and continuous) was calculated for each cluster by averaging the continuous polar angle maps across all individuals within each cluster.

The online version of this article includes the following figure supplement(s) for figure 4:

**Figure supplement 1.** Clusters of eccentricity maps of the dorsal portion of early visual cortex.

**Figure supplement 2.** Average polar angle, eccentricity, and normalized mean BOLD signal maps from each cluster and hemisphere.

indicating that dorsal V3 does not cover the entire quarter visual field (i.e., from 360° to 270°) either throughout its length or only in its most anterior portion. Finally, cluster 6 reflects unclear retinotopic organization, with a handful of individuals' retinotopic maps showing overall low correspondence with the canonical retinotopic organization.

Qualitatively, individual maps seem to agree with their corresponding average cluster map, but there are some exceptions (*Figure 5*, *Figure 5—figure supplements 1–6*). *Figure 5* shows the average cluster maps from each cluster and examples of individuals' maps that are qualitatively similar and dissimilar to their corresponding average cluster map. While most polar angle maps correspond well with their average cluster maps (as seen in the middle row of *Figure 5*), there is also an apparent mismatch between a few maps and their corresponding cluster average (bottom row in *Figure 5*). For example, individual #132118 was assigned to cluster 4, but their polar angle map is qualitatively more similar to cluster 5. These mismatches are likely due to the extensive overlap between within- and between-cluster distributions of pair-wise Jaccard scores (*Figure 6*). Note in *Figure 6* that the within-cluster distributions highlighted in gray are generally shifted to the right compared to the

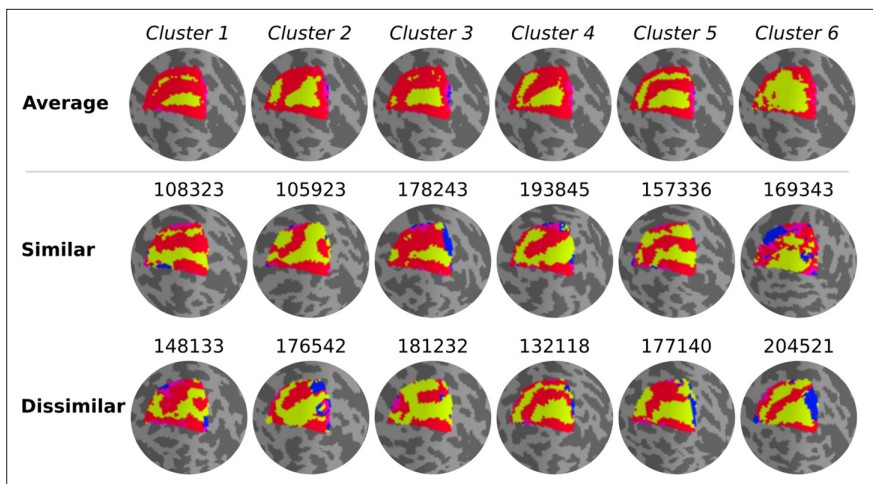

**Figure 5.** Qualitative evaluation of clusters. Average cluster maps are shown in the top row. The middle row shows examples of maps from each cluster with a similar retinotopic organization to the corresponding average map. Finally, in the bottom row, examples of those with dissimilar organizations are shown.

The online version of this article includes the following figure supplement(s) for figure 5:

**Figure supplement 1.** Nine randomly selected polar angle maps within cluster 1.

**Figure supplement 2.** Individual polar angle maps within cluster 2.

**Figure supplement 3.** Nine randomly selected polar angle maps within cluster 3.

**Figure supplement 4.** Nine randomly selected polar angle maps within cluster 4.

**Figure supplement 5.** Nine randomly selected polar angle maps within cluster 5.

**Figure supplement 6.** Nine randomly selected polar angle maps within cluster 6.

between-cluster distributions, indicating their higher Jaccard scores. However, the overlap between these distributions is substantial. For example, the between cluster 1 and 5 distribution overlaps with within-cluster 1 distribution throughout its entirety, which is justified by the significant similarity between their average maps. Despite this, we found that the average within-cluster Jaccard score is 0.54 (standard deviation (SD) = 0.07), while the average between-cluster score is 0.46 (SD = 0.08), showing that pairs of maps within a cluster are, on average, more similar than between clusters.

## Visual field sign analysis for delineating visual area boundaries

Finally, we further examined retinotopic maps with a Y-shaped lower vertical representation, that is, those primarily assigned to cluster 2, to elucidate the kind of deviation from the canonical maps they represent (*Figure 7* and *Figure 7—figure supplement 1*). To do so, we performed a visual field sign analysis (*Sereno et al., 1995*; *Sereno et al., 1994*), which combines both polar angle and eccentricity maps into a unique representation of the visual field as either a non-mirror-image or a mirror-image representation of the retina (*Figure 7a*; see Materials and methods). With such representation, we can directly infer visual area parcellation.

*Figure 7b* shows polar angle gradients in a 'streamline' representation and their respective visual field sign representation for two participants with canonical and four with Y-shaped lower vertical representations. While visual area boundaries in early visual cortex are conventionally identified by reversals in the progression of the polar angle values – or changes in the direction of polar angle gradients, it is unclear how to delineate boundaries in the dorsal portion of polar angle maps in those participants with non-canonical maps (note that their respective ventral portion followed the classical representation), but not on those with canonical maps. However, with the visual field sign representation, the boundaries delineating dorsal V2 in those participants with non-canonical maps are more explicit, and it reveals that the area identified as dorsal V3 shows a discontinuity in the expected mirror-image representation. Such representation has been proposed as the 'incomplete-V3' model of the third-tier cortex for the macaque (*Angelucci and Rosa, 2015*) and other similar models for the owl monkey (*Sereno et al., 2015*) and the marmoset monkey (*Rosa and Tweedale,*

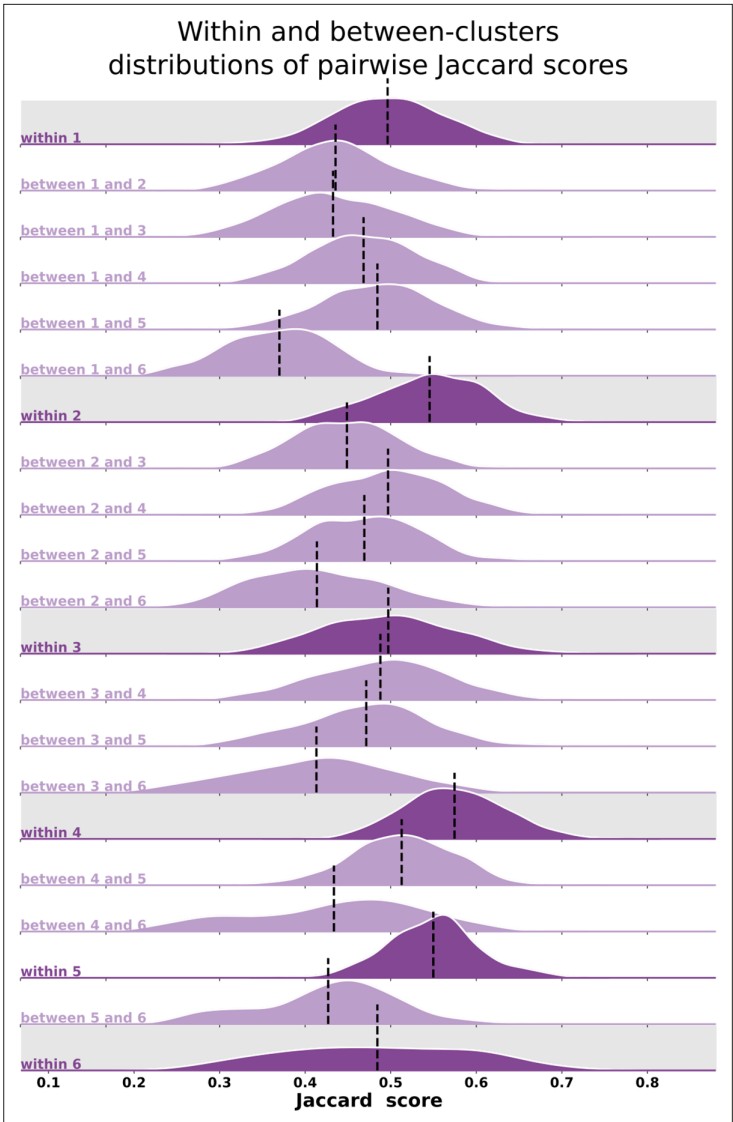

**Figure 6.** Distributions of pair-wise Jaccard scores. Within- and between-cluster distribution of Jaccard scores across all pairs of individuals. Within-cluster distributions are highlighted in gray. Between-cluster distributions are the same regardless of the order of the clusters, that is, the Jaccard score distribution between clusters 1 and 2 (between 1 and 2) is the same as the one between clusters 2 and 1. Black vertical lines indicate distributions' means.

*2000*). *Figure 7—figure supplement 1* shows five more examples of polar angle maps with unusual Y-shaped lower vertical representations and five other examples with a truncated V3 boundary. While individuals with unusual Y-shaped lower vertical representations have a discontinuity in the canonical mirror-image representation of the retina in dorsal V3, individuals with a truncated V3 do not show such a discontinuity. Altogether, our findings may suggest that, at least in humans, the canonical model does not oppose other models established for non-human primates; these models coexist and reflect common modes of variability in the retinotopic mapping of early visual areas.

## Discussion

In this study, we systematically investigated individual variability in visual field representation of human early visual cortex using the HCP 7T retinotopy dataset. We found that retinotopic maps in the left hemisphere were more variable than those in the right hemisphere. Moreover, in the left hemisphere the dorsal portions of early visual areas were more variable than their ventral counterparts, and these

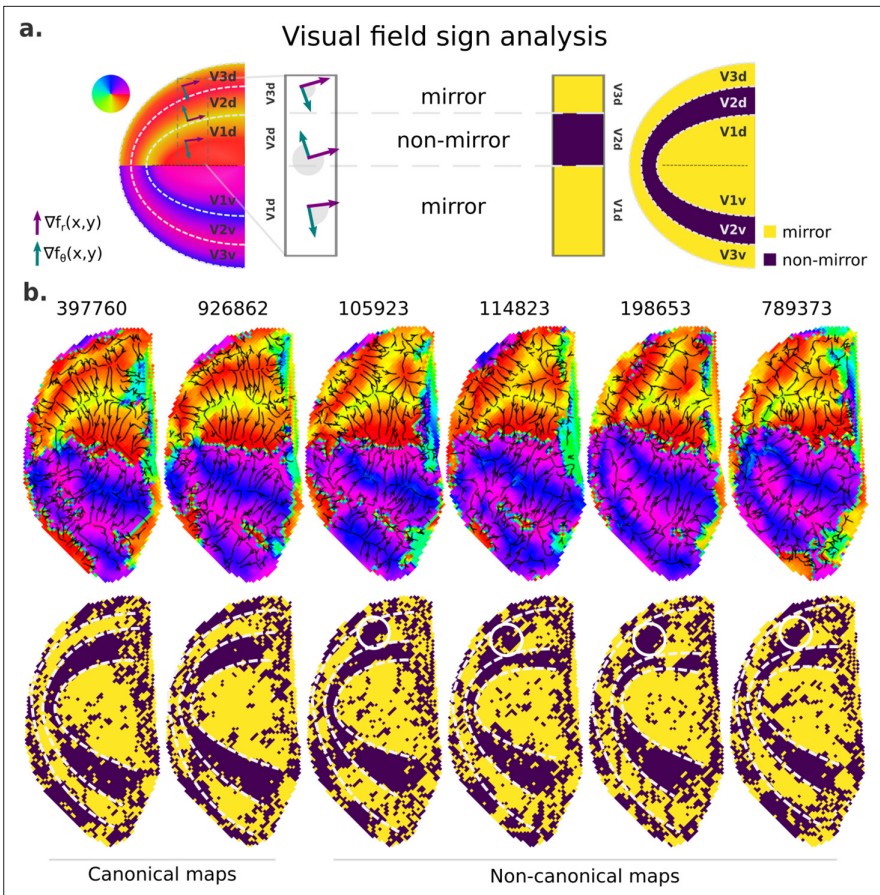

**Figure 7.** Visual field sign analysis for delineating visual areas. (**a**) The visual field sign analysis (*Sereno et al., 1995*; *Sereno et al., 1994*) combines polar angle and eccentricity maps (not shown) into a unique representation of the visual field as either a non-mirror-image (like V2) or a mirror-image (like V1) representation of the retina. This analysis consists of determining the angle between the polar angle and eccentricity maps' gradient vectors, respectively, the green and purple vectors, at each cortical coordinate. If the angle between the gradient vectors is between 0 and $\pi$, by convention, the cortical patch is a mirror-image representation of the retina; otherwise, it is a non-mirror-image. (**b**) Six examples of left hemisphere polar angle maps with canonical (on the left) and non-canonical (on the right) representations in the dorsal portion of early visual cortex are shown (top row). Polar angle gradients are shown in a 'streamline' representation to highlight reversals in the progression of the polar angle values. Their respective visual field sign representation (bottom row) is also shown. While it was unclear how to delineate boundaries in the dorsal portion of polar angle maps in those participants with non-canonical maps, the visual field sign representation reveals that the area identified as dorsal V3 shows a discontinuity in the canonical mirror-image representation (solid white circles).

The online version of this article includes the following figure supplement(s) for figure 7:

**Figure supplement 1.** Visual field sign analysis for delineating visual areas.

effects were not merely due to trivial large-scale intra-individual variability in pRF estimates or individual variability in curvature and the mean BOLD signal. Thus, we investigated whether there were common motifs in the observed individual variability in retinotopic maps of left hemispheres. This analysis showed that deviations from the canonical model of continuous, alternating bands of vertical and horizontal meridian representation in V2 and V3 exist in the majority of individuals. Specifically, the visual field sign analysis revealed that the area identified as dorsal V3 shows a discontinuity in the expected mirror-image representation of the retina in individuals with a Y-shaped lower vertical representation in the dorsal portion of early visual cortex. Overall, our findings challenge the current view that the dorsal portions of early visual areas form retinotopic maps which are consistent between individuals and are roughly mirror images of their ventral counterparts.

## Individual variability in retinotopy

Although previous evidence for the variability seen across dorsal early visual cortex in humans has been mostly anecdotal, a number of studies have indicated a complex, retinotopic organization of dorsal early visual areas in non-human primates, using both electrophysiological recordings and high-resolution fMRI (*Angelucci and Rosa, 2015*; *Gattass et al., 1988*; *Sereno et al., 2015*; *Zhu and Vanduffel, 2019*). Accordingly, there is a long-standing debate about the number of visual areas – and their boundaries – in the third-tier visual cortex of New and Old-World monkeys (*Angelucci and Rosa, 2015*; *Hadjidimitrakis et al., 2019*). However, the question of whether the areal boundaries in this region show significant individual variability has not been studied systematically in non-human primates. Only (*Gattass et al., 1988*) reported, in the macaque monkey, that the representation of the lower vertical meridian in dorsal V3 varied across individuals, but firm conclusions could not be drawn due to the small sample. These authors indicated that some animals showed a continuous representation of this meridian along the rostral border of this area, whereas in others additional field discontinuities created a discontinuous representation. Notably, the same discontinuities in the anterior border of dorsal V3 were also found in our systematic investigation of individual variability in human polar angle maps. It is also significant that the same pattern of variation (relatively simple and reproducible representations of the upper contralateral quadrant, and complex and variable representations of the lower quadrant) characterize V2 and V3 in at least one non-primate, the cat (*Rosa and Manger, 2005*; *Tusa et al., 1979*). Overall, our findings in humans demonstrate that the organization of dorsal early visual areas is more heterogeneous than previously acknowledged and suggest that this may be a common feature of mammals with developed vision.

Our results also indicate that the variability in retinotopic organization increases between V1 and V2, and between V2 and V3. These findings are compatible with a model whereby retinotopic maps develop sequentially from 'core' areas like V1, where maps are determined by genetically encoded molecular gradients, toward late-maturing areas, where their formation occurs interactively through a wire-length minimization rule, allowing progressively greater degrees of freedom (*Yu et al., 2020*). The fact that the organization of ventral maps tends to be more reproducible than that of dorsal maps may be related to the presence of the middle temporal area (MT, or V5) as a second 'anchor' node for the formation of dorsal maps in the proximity of the dorsal cortex, resulting in merging maturational gradients in the region of dorsal V3 and V4 (*Rosa and Tweedale, 2005*).

Although different models of third-tier visual cortex organization in non-human primates (*Angelucci and Rosa, 2015*) also suggest unusual eccentricity mapping, we did not find meaningful differences in clusters of eccentricity maps (*Figure 4—figure supplement 1*) nor in the average eccentricity maps based on clusters from *Figure 4c* (*Figure 4—figure supplement 2*). This may be associated with the limited extent of the visual stimulus (up to 8° of eccentricity) (*Benson et al., 2018*) and remains to be further investigated. Indeed, much of the controversy regarding the organization of dorsal areas in non-human primates refers to regions anterior to the peripheral representations in V2 (*Angelucci and Rosa, 2015*). Another alternative is having a complex pattern of polar angle representation coexisting with a preserved eccentricity gradient, as demonstrated by previous work in areas V2 and V3 of cats (*Tusa et al., 1979*), flying foxes (*Rosa, 1999*), ferrets (*Manger et al., 2002*), and tree shrews (*Sedigh-Sarvestani et al., 2021*).

Our investigation provides firm evidence for individual variability in the retinotopic organization across parts of early visual areas in the human visual cortex. Moreover, the exploratory analysis indicates the presence of shared patterns of retinotopic organization that deviate from the canonical polar angle representation in the dorsal portion of early visual cortex. Future work could extend these insights through additional analyses – for example, by employing different similarity metrics, using different features, or changing the number of clusters. Here, we limited our analysis to the spatial overlap of discrete polar angle maps, which means that a pair of qualitatively similar but spatially misaligned polar angle maps, for example, might have a low Jaccard score. If another more suitable metric can consider the topographic organization of polar angle maps regardless of the spatial location, it would be possible to increase the consistency between an individual's map and their cluster average map. It would also be possible to estimate the similarity between two individuals' retinotopic maps from specific features extracted from the maps, such as linear magnification along isoeccentricity lines, to provide insights into changes in these properties as a function of cortical location (*Schira et al., 2010*). Another option would be to use the visual field sign representation directly for

clustering individuals based on the non-mirror-image and mirror-image representation of the retina across early visual cortex. Finally, it is important to note that selecting the ideal number of clusters depends on the similarity metric employed, prior knowledge, and the clustering algorithm. Therefore, future work could be performed to explore the effect of the number of clusters on clustering quality (perhaps as indicated by within- vs. between-cluster similarity measures).

## Potential sources of individual variability in retinotopy

Given the presence of individual variability across early visual cortex in humans, another potential line of investigation involves the origin of this variability. Pertinently, we recently developed a deep learning model of retinotopy able to predict this individual variability from anatomical information (including curvature) (*Ribeiro et al., 2021*), suggesting that it is, to some extent, a structure-related variation. Accordingly, here we modeled individual variability in curvature as an anatomy-related covariate, which significantly affected the individual variability of polar angle maps. Additionally, to determine if our findings were not merely due to other covariates, we included the normalized mean BOLD signal and intra-individual variability in pRF estimates as additional covariates. The former is an important proxy for the location of large veins (*Boyd Taylor et al., 2019*; *Kurzawski et al., 2022*), which are known to affect pRF estimates (*Boyd Taylor et al., 2019*; *Winawer et al., 2010*). The latter is a proxy for the reliability of individuals' retinotopic maps, which could vary across visual areas. While we found a significant effect of intra-individual variability, the main effects of all factors (hemispheres, visual areas, and portions) on individual variability of polar angle maps persisted. This finding indicates that these large-scale effects were not merely due to the covariates included in the model. Note, however, that as we only consider the effect of large-scale deviations (averaged over a region of interest) of each of these covariates on large-scale variability of the retinotopic maps, it could be the case they still have a crucial role in local variability, which we further discuss in the Limitations.

Retinotopic maps could also vary as a function of data resolution and cortical depth. For example, signal-to-noise ratio (SNR) and partial volume artifacts are directly affected by data resolution (or voxel size); that is, both reduce with voxel size (*Hoffmann et al., 2009*). While lower SNR might lead to noisier (or less smooth) retinotopic maps (*Hoffmann et al., 2009*), the gain from the reduced susceptibility to partial volume artifacts will likely result in increased validity of the observed maps. Partial volume artifacts may arise from patches of opposite walls of a sulcus running across a single voxel or even small vessels, leading to inaccurate signals from the combination of different brain regions and tissues. With increasing magnetic field strength, it should be easier to strike the right balance between high SNR and low partial volume artifacts, which is crucial for determining the impact of registration errors due to partial volume artifacts on the variability of retinotopic maps. Moreover, previous studies have also shown that the hemodynamic response function (*Puckett et al., 2016*) and a spatial pattern of activation (*Polimeni et al., 2010*) varied across depths in V1. Specifically, Polimeni et al. found that a spatial pattern of activation (an 'M') becomes clearer from the white matter surface to the mid-thickness surfaces and then deteriorates once again near the pial surface. Altogether, these studies motivate a more thorough investigation of how retinotopy, as measured by fMRI, varies as a function of data resolution and cortical depth and its implication on individual variability in retinotopy. However, it is also important to note that studies of the columnar organization of non-human primates using single-cell recordings have not found any evidence that the receptive field location varies with cortical depth, although the receptive field size changes, being smallest in the middle layers (*Hubel and Wiesel, 1974*; *Rosa et al., 1997*).

Another potential research direction is determining the extent to which eye movement underlies some of the variability found in retinotopic maps of early visual areas. For example, one could systematically evaluate pRF modeling accuracy as a function of gaze position change (and other eye-tracking signal derivatives). We performed a preliminary analysis of the deviation in gaze position at each time point and averaged across runs of retinotopic mapping stimuli and individuals assigned to each cluster. *Supplementary file 6* summarizes the average deviation of gaze position from the fixation point along the *X* and *Y* axes for each cluster. In brief, we did not find consistent results across clusters. However, we do not rule out the effect of eye movement on the variability of the retinotopic maps because the eye-tracking data quality is variable and unavailable for some individuals in the HCP retinotopy dataset (*Benson et al., 2018*), and the clustering quality could be improved (e.g., through manual clustering or the other approaches previously discussed). Another possibility is determining

the reliability of these retinotopic maps through connective field modeling (*Haak et al., 2013*) with unconstrained eye movement data (*Tangtartharakul et al., 2023*). Yet, we observe that unusual maps in the dorsal portion of the early visual cortex coincide with canonical representations in the ventral portion (*Figure 7*, *Figure 7—figure supplement 1*, and *Figure 4—figure supplement 2*), which is unlikely to be the case for noisy data driven by massive eye movements.

## Limitations

Although we demonstrate that common modes of deviation from the canonical dorsal V2 and V3 organization exist in the left hemisphere, further analyses are necessary to fully ascertain if such variability is indeed neurogenic or if it could be a result of measurement errors. Among potential sources of measurement error are the presence of large veins running adjacent to regions of interest. Accordingly, by using the normalized mean BOLD signal as a proxy for the location of large veins (*Boyd Taylor et al., 2019*; *Kurzawski et al., 2022*), studies have shown that voxels near these veins show lower mean BOLD signal, a phenomenon known as the venous eclipse, which affects pRF estimates, for example, in area hV4 (*Boyd Taylor et al., 2019*; *Winawer et al., 2010*). As such, deviations in the expected mean BOLD signal could result in deviations from the expected retinotopic organization. Thus, to better understand the potential effects of the presence of large veins on the different levels of variability in visual field representation across early visual areas, we considered both the pair-wise correlations between retinotopic maps and the normalized mean BOLD signal (*Figure 3*) and the large-scale deviation in the normalized mean BOLD signal as covariates in the LME model (*Tables 3 and 4*).

In the pair-wise correlation analysis, we found that the polar angle representation is indeed correlated with the normalized mean BOLD signal, and the magnitude of such association varies across visual areas. Specifically, we found a higher correlation (in magnitude) between the polar angle and normalized BOLD signal in ventral visual areas than in dorsal areas. This difference could be explained by ventral areas' proximity to the dural venous sinuses (*Winawer et al., 2010*), that is, the transverse sinus, the superior sagittal sinus, and the straight sinus. These venous sinuses are known to introduce artifacts to the BOLD signal, which might lead to changes in retinotopic maps. Importantly, none of these sinuses run near the dorsal V2 and V3 (see *Figure 4—figure supplement 2*, in which it is possible to observe the likely confluence of sinuses), and both of which show the lowest correlation between polar angle and normalized BOLD signal maps.

Moreover, we modeled large-scale deviation in visual field maps using the large-scale deviation of the normalized mean BOLD signal as a covariate. We did find a significant effect of the normalized mean BOLD signal on the individual variability of eccentricity (*Table 4*) but not of polar angle maps (*Table 3*). Thus, to determine the effectiveness of the LME model for uncovering the effects of covariates, we similarly modeled the large-scale variability in polar angle maps of hV4 (not shown in the manuscript). We did not find an effect of the normalized mean BOLD signal on large-scale deviation in polar angle maps in hV4. However, we found a weak correlation between the polar angle and normalized mean BOLD signal (LH: $r = -0.06$; RH: $r = -0.07$). This finding does not reconcile with previous reports, that is, that venous artifact impacts retinotopy. However, a fine-grained analysis of polar angle maps in hV4 indicated the inconsistent effect of the venous artifact on polar angle mapping (*Boyd Taylor et al., 2019*). In our analysis, though, these results might reflect an inability to appropriately parcellate hV4 at the individual level when using an atlas-based parcellation; while early visual areas are more consistently found at specific spatial locations, this is not the case for other visual areas of which spatial location and extent seem to vary across participants. Therefore, although we did not find a significant effect of the normalized mean BOLD signal in our LME model, it does not mean that the macro- and microvasculature do not affect retinotopy. For the former, future research might consider a fine-grained analysis of topographic deviations, such as the one reported by (*Boyd Taylor et al., 2019*). For the latter, given that the mean BOLD signal is only used as a proxy for the location of large veins, a more detailed analysis of the microvasculature might require other imaging data, such as high-resolution time-of-flight magnetic resonance angiography data (*Bollmann et al., 2022*).

## Future directions

Our findings raise questions about if and how cortical atlases should be revisited to accommodate deviations from the canonical model of retinotopic organization, especially for dorsal V3. Here, by

using the visual field sign representation, we could better understand what kind of deviation from the canonical model atypical maps represent. Therefore, using such data representation could be helpful for the manual segmentation of atypical maps. Alternatively, combining deep learning models (*Ribeiro et al., 2021*) for generating a retinotopic prior that accommodates more variability with a Bayesian framework (*Benson and Winawer, 2018*) for boundary delineation might prove fruitful to support the need for automated individual-level parcellation methods. It would also be desirable to use functional characteristics of areas, such as the responses to specific types of visual stimulation, to increase the confidence in assignment of boundaries. Whereas this may be possible in some situations (e.g., using motion selectivity as a localizer for area MT; *Pitzalis et al., 2010*), attempts to segregate V3 from adjacent areas on this basis may be more challenging, due to the physiological similarity between this area and the adjacent V2 and V3a (*Gegenfurtner et al., 1997*; *Levitt et al., 1994*; *Zeki, 1978*). Differences in pRF size (*Zhu and Vanduffel, 2019*) could offer some insight, although the wide overlap in the distributions of single-unit receptive field sizes in adjacent areas (*Rosa, 1997*) suggests that obtaining clear-cut boundaries on this basis remains unlikely.

Importantly, accommodating individual variability for the automatic parcellation of visual areas is also crucial for understanding the functional properties of the human visual cortex. These analyses require the precise delineation of boundaries between visual areas, either by manually tracing transitions in visual field maps or by using an automatic segmentation method (*Benson et al., 2014*; *Benson and Winawer, 2018*; *Dougherty et al., 2003*). In both cases, a spatially consistent mapping (i.e., a canonical representation) of continuous, alternating bands of vertical and horizontal meridian representation in V2 and V3 is often assumed. However, we demonstrate that deviations from the canonical model exist, especially in the left hemisphere and dorsal V3. This finding could have several implications for post hoc analyses requiring visual area delineation, suggesting that previous studies may have mischaracterized differences between these visual areas due to misidentification. Therefore, an important future direction of this work is determining how functional selectivity varies across visual areas parcellated according to the canonical model compared to using new parcels estimated with the visual field sign analysis.

In addition, the present findings highlight the need for a more comprehensive assessment of the degree of variability in visuotopic maps in non-human primates, where a higher degree of precision can be achieved with invasive methods including single neuron recordings and optical imaging of intrinsic signals. To date, variability has only been reported in macaque monkeys (*Gattass et al., 1988*), but the available data in marmoset and owl monkeys indicate a reproducible organization that does not fully agree with the canonical model of dorsal V3 (*Angelucci and Rosa, 2015*; *Rosa and Schmid, 1995*; *Rosa and Tweedale, 2000*; *Sereno et al., 2015*). Whether this simply reflects the small number of individuals explored, or a truly more stable configuration (perhaps associated with the larger brain size in humans; *Angelucci and Rosa, 2015*; *Rosa and Tweedale, 2005*) remains to be determined.

Another finding that requires consideration is the interhemispheric difference revealed in our data: retinotopic maps in the left hemisphere showed more variation than those in the right hemisphere. To date, there has been no report of interhemispheric differences in early visual cortex of other mammals, including non-human primates. In part, this may be traced to the relatively small samples in these studies compared to those possible using human fMRI. However, another possibility is that such differences may arise more frequently in human brains due to the scaling of callosal connections with brain size (*Rilling and Insel, 1999*), which may promote a higher degree of connectional independence during development.

Finally, these results raise many questions regarding how retinotopic maps develop. For example, studies modeling the formation of retinotopic maps in development have suggested that multistable solutions may occur depending on factors such as the degree of elongation of the area (*Sedigh-Sarvestani et al., 2021*; *Wolf et al., 1994*) and adjacency with other areas (*Yu et al., 2020*), which do not violate the need to minimize the length of connections (*Durbin and Mitchison, 1990*; *Swindale, 1996*). Therefore, future work could evaluate whether there is an overlap between function- and anatomy-based clusters to help elucidate the developmental mechanisms underlying the variability of human dorsal extrastriate cortex.

In conclusion, using a large-scale brain imaging dataset, we provide new insights into the variability in the topographical organization of human visual cortex. These insights may prove crucial in guiding

further experimental investigations and theories about retinotopic organization differentiation across species, development, and individuals.

## Materials and methods

### Dataset

We used the HCP 7T Retinotopy dataset (*Benson et al., 2018*) to investigate individual variability in retinotopic maps of human early visual cortex. This dataset consists of high-resolution functional retinotopic mapping and structural data from 181 participants (109 females, age 22–35) with normal or corrected-to-normal visual acuity. Participant recruitment and data collection were led by Washington University and the University of Minnesota. The Institutional Review Board (IRB) at Washington University approved all experimental procedures (IRB number 201204036; 'Mapping the Human Connectome: Structure, Function, and Heritability'), and all participants provided written informed consent before data collection (*Van Essen et al., 2013*). Additionally, the acquisition protocol has been described in previous work (*Benson et al., 2018*; *Van Essen et al., 2013*).

Structural data were acquired at 0.7-mm isotropic resolution in a customized Siemens 3T Connectome scanner (*Van Essen et al., 2013*). Briefly, cortical surfaces were reconstructed from T1w structural images using FreeSurfer and aligned to the 32k fs_LR standard surface space. This standard 32k fs_LR cortical surface consists of 32,492 vertices sparsely connected, forming triangular faces. Functional data were later aligned with this standard surface space.

Functional retinotopic mapping data were acquired using a Siemens 7T Magnetom scanner at 1.6-mm isotropic resolution and 1-s TR. Data were preprocessed following the HCP pipeline (*Glasser et al., 2013*), which included correction for head motion and EPI spatial distortion, alignment of the fMRI data with the HCP standard surface space, and denoising for spatially specific structured noise. Retinotopic mapping stimuli comprised rotating wedges, expanding and contracting rings, and bars of different orientations moving across different directions in the visual field. A population receptive field (pRF) modeling procedure was then used to reconstruct visual field maps (*Benson et al., 2018*; *Dumoulin and Wandell, 2008*; *Kay et al., 2013*), which encompasses estimating the spatial preference of cortical surface vertices to different locations of the visual field (i.e., its receptive field) defined in polar coordinates – for more, see *Benson et al., 2018*. Hence, polar angle maps are retinotopic maps reflecting the polar angle (angle relative to the horizontal vertical meridian) in the visual field to which a vertex is most responsive, while eccentricity maps reflect the distance from the center of the visual field (i.e., the fixation point). The combination of a polar angle map and an eccentricity map completely specifies a map of the visual field, from the center of the visual field to at least 8° of eccentricity (*Benson et al., 2018*).

### Region of interest

Early visual areas were defined by a surface-based probabilistic atlas (*Wang et al., 2015*). This probabilistic atlas includes the dorsal and ventral portions of V1, V2, and V3, not including the foveal confluence. For the clustering analysis, we slightly modified the atlas by extending the dorsal border of V3 and including V1/V2/V3 foveal confluence (*Schira et al., 2009*), in line with our previous work (*Ribeiro et al., 2021*).

### Individual variability

We determined individual variability in visual field maps to quantify how variable these maps were across visual areas (V1, V2, and V3), portions (dorsal and ventral), and hemispheres (left and right) in human early visual cortex. First, we computed the average retinotopic maps across all 181 individuals from the HCP retinotopy dataset for both left and right hemispheres (*Figure 1—figure supplement 1*). Then, we iteratively calculated the vertex-wise difference between an individual's retinotopic map and the average map. The difference between two angles is given by:

$$\text{MIN}(|\hat{\theta} - \theta|, |\hat{\theta} - \theta + 2\pi|, |\hat{\theta} - \theta - 2\pi|) \tag{1}$$

for $0 < \theta < 2\pi$.

Finally, vertex-wise difference scores were averaged over vertices in the range of 1–8° of eccentricity within the dorsal and ventral portions of early visual areas, resulting in one scalar value per individual

per visual area, which we refer to as the individual variability. The eccentricity mask was defined using the group-average eccentricity map. This range of eccentricity values was chosen because, in the original population receptive field mapping experiment of the HCP, the visual stimulus extended to 8° of eccentricity (**Benson et al., 2018**). Additionally, due to the inherent difficulty in mapping the foveal confluence (**Schira et al., 2009**), we constrained our comparison to eccentricity values above 1°. According to studies in non-human primates, this corresponds approximately to half of the expected extent of V1, V2, and V3 (**Gattass et al., 1988**; **Gattass et al., 1981**).

## LME model

We determined whether there were main effects and interactions of hemispheres (left and right), visual areas (V1, V2, and V3), and portions (dorsal and ventral) on individual variability of retinotopic maps using LME models. Standard analyses of variance and *t*-tests assume statistical independence of individuals' data (**Yu et al., 2022**), which is often not the case. For example, the 7T HCP retinotopy dataset includes data from 50 monozygotic and 34 dizygotic twins, totaling 168 individuals out of 181. Therefore, to meet the statistical independence criterion, many data points would have to be disregarded for standard statistical inference. However, LME models allow us to take full advantage of the dataset by explicitly modeling cluster-specific means (random intercepts). Indeed, individual variability from different visual areas is naturally clustered by individuals (**Magezi, 2015**). Therefore, using this statistical model, we can appropriately model individual-specific effects (**Magezi, 2015**; **Yu et al., 2022**).

In our LME model, the dependent variable is the individual variability ($Y$), which is modeled as a function of the fixed effects ($\beta$) of three factors ($x$) and their interactions. These three factors are: hemisphere, visual area, and portion. Additionally, we also consider the random effects ($\gamma_i$) associated with the individual ($i = 1, \ldots, 181$), and the random effects of each factor nested within the individual ($\gamma_{ij}$, with $j = 1, 2$, and 3). This model is expressed as:

$$Y_i = \beta_0 + \sum_{j=1}^{3} \beta_j x_j + \beta_{12} x_1 x_2 + \beta_{13} x_1 x_3 + \beta_{23} x_2 x_3 + \beta_{123} x_1 x_2 x_3 + \sum_{j=1}^{3} \gamma_{ij} + \gamma_i + \varepsilon_i \tag{2}$$

where $\beta_0$ is the intercept and $\varepsilon$ is the residual random error. We built two separate models for individual variability associated with polar angle and eccentricity maps using Jamovi (**The Jamovi project, 2021**, n.d.).

## Individual variability in retinotopic maps and potential confounds

Other potential sources of individual variability in retinotopic maps can be readily included as covariates in LME models after their appropriate standardization by computing *z*-scores. Effectively, adding a covariate to an LME model adds a $\beta x$ term to **equation 2**, where $\beta$ is the estimated effect and $x$ is the covariate. Here, we consider the effect of potential confounders, including curvature, normalized mean BOLD signal, and intra-individual variability in pRF estimates. These factors are known to be correlated with retinotopy and could explain part of the variability found across visual areas. Accordingly, we calculated pair-wise correlations among polar angle, eccentricity, curvature, and normalized mean BOLD signal for both V1–3 and each visual area separately. Given a region of interest, each topographic map was vectorized, and data were concatenated across all participants ($n = 181$) for each modality. Polar angle maps were converted such that 0° corresponds to the horizontal meridian and 90° corresponds to the upper and lower vertical meridians (**Kurzawski et al., 2022**). Finally, pair-wise correlations were determined using these concatenated sets of vectorized maps. Below we provide further motivation for considering each of these covariates.

Curvature maps highlight the geometry of folding patterns of individuals' cortical surfaces, with negative values representing sulci and positive gyri (as per the HCP dataset). In V1, the horizontal and vertical meridian representations of the visual field correlate with underlying sulcal patterns (**Hinds et al., 2008**; **Holmes and Lister, 1916**; **Horton and Hoyt, 1991**; **Inouye, 1909**; **Rajimehr and Tootell, 2009**). This tight structure–function relationship was further leveraged for individual-level predictions of retinotopy from underlying anatomy, using atlas-fitting algorithms (**Benson et al., 2014**; **Benson et al., 2012**), a Bayesian model (**Benson and Winawer, 2018**), and deep learning (**Ribeiro et al., 2021**). Therefore, although curvature is a good predictor of retinotopy, it is unclear if it is sufficient to

explain the systematic differences in individual variability across visual areas. Here, we consider the individual variability in curvature maps as a covariate. Individual variability in curvature was operationalized as the mean absolute difference between an individual's curvature map and the corresponding average map over vertices in the range of 1–8° of eccentricity within the dorsal and ventral portions of early visual areas, resulting in one scalar value per individual per visual area.

The mean preprocessed BOLD signal was used as a proxy for the location of large veins (**Boyd Taylor et al., 2019**; **Kurzawski et al., 2022**), which are known to affect pRF estimates. Voxels near large veins show lower mean BOLD signal, a phenomenon known as the venous eclipse, which affects pRF estimates, for example, in area hV4 (**Boyd Taylor et al., 2019**; **Winawer et al., 2010**). As such, deviations in the expected mean BOLD signal could lead to deviations from the expected retinotopic organization. Therefore, we considered individual variability in the mean BOLD signal as another covariate. To do so, we first normalized the mean BOLD signal (over the time course) by dividing the value of each vertex by the maximum intensity (**Boyd Taylor et al., 2019**). Then, we computed the individual variability as previously described for curvature maps.

Finally, different levels of individual variability in retinotopic maps could also reflect variations in their reliability across visual areas. For example, in the HCP dataset, despite pRF estimates being highly consistent between two model fits (**Benson et al., 2018**), there was still some intra-individual (between-fit) variability. Thus, we considered intra-individual variability in pRF estimates as a covariate. Intra-individual variability was operationalized as the difference in pRF estimates from two pRF model fits of half splits of the retinotopic mapping data (fit 2 and 3; **Supplementary file 7**), which were provided and determined using the first and second half of the temporal data from each run (**Benson et al., 2018**). By using these two other fits, it is possible to assess the reliability of the pRF parameter estimates; hence, the difference in pRF estimates is a proxy for the reliability of individuals' retinotopic maps and can be included as a covariate in the LME model.

## Clusters of spatial organization

We performed an exploratory clustering analysis to determine whether retinotopic maps differ from the average map in similar ways, particularly in the dorsal portion of early visual cortex. Specifically, we investigated the spatial overlap between retinotopic maps as an unambiguous indicator of the similarity between two maps. First, to obtain such a measure of the spatial overlap, the continuous polar angle maps were converted into discrete maps, such that each vertex was categorized into one out of four possible labels:

$$\theta_{discrete} = \begin{cases} 0°, for\ 0° \leq \theta_{continuous} \leq 45° \\ 90°, for\ 45° < \theta_{continuous} \leq 180° \\ 270°, for\ 180° \leq \theta_{continuous} < 315° \\ 360°, for\ 315° \leq \theta_{continuous} < 360° \end{cases}$$

these categories were chosen because they highlight the location of visual area boundaries. Discrete eccentricity maps were determined by:

$$\theta_{discrete} = \begin{cases} 0°, for\ 0° \leq \theta_{continuous} \leq 2° \\ 2°, for\ 2° < \theta_{continuous} \leq 4° \\ 4°, for\ 4° < \theta_{continuous} \leq 6° \\ 6°, for\ 6° < \theta_{continuous} \end{cases}$$

Next, the spatial overlap between discrete maps from all possible pairs of individuals was estimated using the Jaccard similarity coefficient (**Levandowsky and Winter, 1971**; **Taha and Hanbury, 2015**). The Jaccard index estimates similarity between two maps by taking the size of the intersection (in number of vertices) divided by the size of the union of two label sets. Hence, the Jaccard score ranges from 0 to 1; the closer to 1 the score is, the more similar the two maps are. For our data and each pair of individuals, the Jaccard index is determined from the two possible individuals' combinations (i.e., individual 1 vs. individual 2 and individual 2 vs. individual 1) since the order of the maps

determines which map is the reference. For each combination, we estimated the Jaccard index for each label, and their weighted average was determined using the number of labels' instances in the reference map to account for label imbalance. Then, these two estimates were averaged, resulting in one estimate of the spatial overlap between two individuals' discrete retinotopic maps.

To assess whether inter-individual differences fell into stereotyped patterns, we applied a spectral clustering algorithm from Scikit-learn (*Abraham et al., 2014*; *Pedregosa et al., 2011*). This algorithm operates on the low-dimensional embedding of the affinity matrix (our Jaccard index-based similarity matrix), followed by *K*-means clustering of the components of the eigenvectors in the low-dimensional space. This low-dimensional space is determined by selecting the most relevant eigenvectors of the graph Laplacian of the affinity matrix, of which corresponding eigenvalues reflect important properties of the affinity matrix that can be used to partition it (*von Luxburg, 2007*). In implementing the spectral clustering algorithm, we set the number of clusters to 6 and fixed the random state for replication purposes. We selected this number of clusters as there are at least five different models of third-tier visual cortex organization in non-human primates (*Angelucci and Rosa, 2015*), with a sixth cluster intended to capture noisy or unclear retinotopic organization. Note, however, that this selection is simply a speculation of the possibility space and do not reflect known inter-individual differences in non-human primates. After clustering, we computed each cluster's mean map by averaging the continuous retinotopic maps across individuals within each cluster.

## Visual field sign analysis

Lastly, we further examined unusual retinotopic maps to elucidate the kind of deviation from the canonical maps they represent. We performed a visual field sign analysis (*Sereno et al., 1995*; *Sereno et al., 1994*), which combines both polar angle and eccentricity maps into a unique representation of the visual field as either a non-mirror-image (like V2) or a mirror-image representation of the retina (like V1). Since the left hemisphere represents the right visual field, which in polar angle includes 0–90° (upper right visual field) and 270–360° (lower right visual field), we shifted the polar angle values so the point of wrap-around (from 360° to 0°) was positioned at the horizontal meridian in the contralateral hemifield, avoiding the discontinuous representation between 360° and 0°. Then, we interpolated the sparse and flattened polar angle and eccentricity maps onto a regular *x*–*y* grid using SciPy (*Virtanen et al., 2020*). Next, we determined the gradient of polar angle and eccentricity maps, mathematically expressed as:

$$\Delta f_\theta = \frac{\partial f_\theta}{\partial x}\hat{i} + \frac{\partial f_\theta}{\partial y}\hat{j} \text{ and } \Delta f_r = \frac{\partial f_r}{\partial x}\hat{i} + \frac{\partial f_r}{\partial y}\hat{j}$$

where $f_\theta$ is the polar angle map and $f_r$ the eccentricity, using the NumPy gradient numerical method (*Harris et al., 2020*). Finally, the angle between the polar angle and eccentricity maps' gradient vectors was determined at each *x*–*y* coordinate. If the angle between the gradient vectors is between 0 and $\pi$, by convention, the cortical patch is a mirror-image representation of the retina; otherwise, it is a non-mirror-image. After binarizing the angle projection, we can conveniently infer borders between visual areas because adjacent areas often have the opposite visual field sign (*Sereno et al., 1995*; but see *Yu et al., 2020* for caveat).

## Acknowledgements

This work was supported by the Australian Research Council (DE180100433 and DP210101042) and National Health and Medical Research Council (APP1194206). Data were provided by the Human Connectome Project, WU-Minn Consortium (Principal Investigators: David Van Essen and Kamil Ugurbil; 1U54MH091657) funded by the 16 NIH Institutes and Centers that support the NIH Blueprint for Neuroscience Research; and by the McDonnell Center for Systems Neuroscience at Washington University. In addition, FLR acknowledges support through the Australian Government Research Training Program Scholarship. We thank the reviewers (Elisha Merriam, Tomas Knapen, and Noah C Benson) for helpful comments on early versions of our manuscript and Mark Schira for helpful discussions.

## Additional information

### Funding

| Funder | Grant reference number | Author |
| --- | --- | --- |
| Australian Research Council | DE180100433 | Alexander Puckett |
| Australian Research Council | DP210101042 | Elizabeth Zavitz Marcello GP Rosa |
| National Health and Medical Research Council | APP1194206 | Marcello GP Rosa |

The funders had no role in study design, data collection, and interpretation, or the decision to submit the work for publication.

### Author contributions

Fernanda Lenita Ribeiro, Conceptualization, Data curation, Software, Formal analysis, Validation, Investigation, Visualization, Methodology, Writing – original draft, Writing – review and editing; Ashley York, Formal analysis, Methodology, Writing – review and editing; Elizabeth Zavitz, Conceptualization, Funding acquisition, Writing – review and editing; Steffen Bollmann, Conceptualization, Writing – review and editing; Marcello GP Rosa, Conceptualization, Resources, Supervision, Funding acquisition, Writing – original draft, Writing – review and editing; Alexander Puckett, Conceptualization, Resources, Supervision, Funding acquisition, Writing – original draft, Project administration, Writing – review and editing

### Author ORCIDs

Fernanda Lenita Ribeiro (ID) https://orcid.org/0000-0002-1620-4193
Ashley York (ID) https://orcid.org/0000-0001-5151-6160
Elizabeth Zavitz (ID) http://orcid.org/0000-0002-6501-2358
Steffen Bollmann (ID) https://orcid.org/0000-0002-2909-0906
Marcello GP Rosa (ID) https://orcid.org/0000-0002-6620-6285
Alexander Puckett (ID) https://orcid.org/0000-0001-5983-397X

### Ethics

We used the Human Connectome Project (HCP) 7T Retinotopy dataset (Benson et al., 2018) to investigate individual variability in retinotopic maps of human early visual cortex. This dataset consists of high-resolution functional retinotopic mapping and structural data from 181 participants (109 females, age 22–35) with normal or corrected-to-normal visual acuity. Participant recruitment and data collection were led by Washington University and the University of Minnesota. The Institutional Review Board (IRB) at Washington University approved all experimental procedures (IRB number 201204036; 'Mapping the Human Connectome: Structure, Function, and Heritability'), and all participants provided written informed consent before data collection (Van Essen et al., 2013).

### Decision letter and Author response

Decision letter https://doi.org/10.7554/eLife.86439.sa1
Author response https://doi.org/10.7554/eLife.86439.sa2

## Additional files

### Supplementary files

• Supplementary file 1. Linear mixed-effect model of individual variability of polar angle maps. This file was exported from Jamovi.

• Supplementary file 2. Linear mixed-effect model of individual variability of eccentricity maps. This file was exported from Jamovi.

• Supplementary file 3. Fixed effects parameter estimates for the linear mixed-effects model of individual variability of polar angle maps using covariates. This file was exported from Jamovi.

• Supplementary file 4. Fixed effects parameter estimates for the linear mixed-effects model of

individual variability of eccentricity maps using covariates. This file was exported from Jamovi.

• Supplementary file 5. Fixed effects parameter estimates for the linear mixed-effects model of individual variability of polar angle maps using covariates, including age, gender, and gray matter volume as additional covariates. This file was exported from Jamovi.

• Supplementary file 6. Gaze position change as a function of cluster assignment. The mean deviation in gaze position along the X and Y axes across runs of retinotopic mapping stimuli presentation and individuals is shown for each cluster. We also show the number of individuals with eye-tracking data per cluster, given that eye-tracking data are not available for all individuals.

• Supplementary file 7. Summary of the data used for the analyses described in the main manuscript.

• MDAR checklist

### Data availability

The data used in this study are publicly available at BALSA. In addition, all accompanying Python and MATLAB source codes are available on GitHub (copy archived at *Ribeiro and York, 2023*). On our GitHub repository, executable code is available on interactive computational notebooks (Jupyter notebooks) that allow anyone to execute interactive plotting functions with dropdown menus to visualize an individual's polar angle and visual field sign maps, given their cluster assignment. Note that our documentation provides instructions for running them on Neurodesk (*Renton et al., 2022*). Finally, the intermediate files for fitting the linear mixed-effect models using Jamovi are available on the Open Science Framework.

The following dataset was generated:

| Author(s) | Year | Dataset title | Dataset URL | Database and Identifier |
|---|---|---|---|---|
| Ribeiro FL | 2023 | Variability of visual field maps in human early extrastriate cortex challenges the canonical model of organization of V2 and V3 | https://osf.io/tdkuj/ | Open Science Framework, tdkuj |

The following previously published dataset was used:

| Author(s) | Year | Dataset title | Dataset URL | Database and Identifier |
|---|---|---|---|---|
| Benson NC, Jamison KW, Arcaro MJ, Glasser MF, Coalson TS, Essen DCV, Yacoub E, Ugurbil K, Winawer J, Kay K, Vu AT | 2018 | The HCP 7T Retinotopy Dataset | https://balsa.wustl.edu/study/show/9Zkk | The Brain Analysis Library of Spatial Maps and Atlases, 9Zkk |

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
