## [Editor Report]

This fundamental work advances our understanding of the retinotopic organization of early visual cortex in humans, and in particular, patterns of variability in the organization of V2 and V3 in normal populations. The evidence supporting the conclusions is compelling, with rigorous retinotopic analysis across a large publicly-available dataset. The work will be of broad interest to vision scientists, as well as those interested in inter-individual variability and neural fingerprinting using fMRI.

---

## [Decision Letter]

**Decision letter after peer review:**

Thank you for submitting your article "Variability of visual field maps in human early extrastriate cortex challenges the canonical model of organization of V2 and V3" for consideration by *eLife*. Your article has been reviewed by 3 peer reviewers, including Elisha Merriam as the Reviewing Editor and Reviewer #1, and the evaluation has been overseen by Chris Baker as the Senior Editor. The following individuals involved in the review of your submission have agreed to reveal their identity: Tomas Knapen (Reviewer #2); Noah C Benson (Reviewer #3).

Essential revisions:

1) Each of the reviewers raised questions about potential explanations for the alternate organization that the authors report. While it is likely not possible to completely address each of the reviewers' concerns with additional analysis, the authors should consider the reviewers' recommendations, and at the very least provide a serious discussion of the points that are not possible to address with further data analysis.

2) Both Reviewers 1 and 2 recommended providing a more systematic and complete visualization of the alternate organization that the authors report. There are a number of ways in which this could be accomplished. The authors should pursue an approach that most effectively documents the alternative pattern that they have discovered and the range of variability in the HCP data.

3) Reviewer 3 asked the authors to perform an examination of the field sign, which may shed light on the data in Figure 4., or perhaps a further analysis of the patch of cortex bounded by the arms of the "dorsal Y" in Cluster 2 would help elucidate the kind of deviation from the traditional maps this cluster represents.

4) There was a general consensus amongst the reviewers that the LME approach that the authors have employed may not be the most appropriate or the most sensitive analysis. For example, it would be good to demonstrate convincingly that including the normalized mean BOLD signal in the LME is in fact capable of detecting the venous eclipse around V4 (See Reviewer 1, point #2). Without such a positive result showing that the analysis can detect vascular artefacts, it is difficult to fully trust the lack of an effect in the early visual cortex.

*Reviewer #1 (Recommendations for the authors):*

This manuscript concerns the organization of the first three cortical visual areas (V1/V2/V3). For decades, the field has assumed an organization in which V2 and V3 are conceived as alternating, mirror-symmetric quarterfield representations emanating from a full hemifield field representation in V1. Here, the authors have described an innovative and (mostly) compelling analysis demonstrating a striking degree of intersubject variability in this organization. Deviations from the canonical organization are not random. Instead, the authors have uncovered a subset of subjects that follow an alternative organization, most prevalent in the dorsal portion of the left hemisphere.

This study comes as a breath of fresh air, a huge relief to somebody who has personally spent many hundreds of hours over the years hand-drawing visual area boundaries on inflated brains. As anybody who has attempted this task can readily attest, there is indeed a large degree of inter-subject variability, and forcing the canonical organization on some hemispheres can involve more than a bit of guesswork. The present study suggests that the glove simply doesn't fit for these subjects.

I have always assumed that the canonical organization is correct and that the frequent deviations that I've observed were due to measurement errors. My biggest criticism of the present study is that the authors have not gone far enough in demonstrating that this isn't the case. In other words, I am not completely convinced that this alternative organization derives from variability in the neural map, rather than from a systematic artifact, either in the MRI measurements (functional or structural) or in one of the many processing stages to intervene between data acquisition and the final map.

Regardless of whether the results shown here are indeed neurogenic, or are somehow the result of measurement error, the implications are far-reaching. Over the years, countless fMRI studies have shown bar graphs quantifying some effect in areas V1/V2/V3, with areas boundaries that were either drawn by hand or estimated with an atlas, conforming to the standard, canonical organization. A major implication of the current results is that these previous studies may have mischaracterized differences between these visual areas. And if that is the case in the early visual cortex, I am sure this is a much larger problem in higher-order cortical areas. So in sum, I think this is a really important set of observations and I am confident that it will have an impact on the field.

1. The manuscript is very clear and well-written and I am inclined to believe the results. But this inclination is really based on simple visual inspection of the maps shown, and not the statistical analyses per se. Given that the I find visual inspection to be the most compelling presentation of the result, the manuscript could be greatly improved by showing more data visually demonstrating the range of maps in the HCP data. This could be a supplement, or a pointer to an online resource showing the retinotopic maps for all subjects and all hemispheres, perhaps categorized into meaningful groups.

2. The authors have not done an especially convincing job of discounting various artifacts that may have contributed to the variability in map structure. The authors included covariates in the LME for curvature, normalized mean BOLD signal, and intra-individual variability in pRF estimates, and then argue that these covariates do not account for the pattern of variability in map organization. But doesn't this logic ask us to accept the null hypothesis that these factors aren't playing a role? "line 367: Our findings indicate that the main effects found here were not a mere reflection of variation in the reliability of the individual maps and other covariates." But in fact, the analysis simply suggests that an effect could not be detected. It would be more convincing if the authors could demonstrate that such effects do occur and do impact retinotopic maps (such as the venous eclipse in V4), but that they aren't playing a role here. Specifically, I am concerned that including the normalized mean BOLD signal in an LME isn't a very sensitive way of testing for the effects of the vasculature. Demonstrating a positive effect in V4 with this approach (where we know all about the impact of a large draining vein) would be a more convincing demonstration of the method, and enable us to interpret the lack of an effect in V2/V3.

3. In addition to the three covariates that the authors considered, I wonder about additional issues, such as segmentation reliability, cortical thickness, partial voluming, and bleed over from the wall of the adjacent sulcal surface (i.e., multiple surfaces passing through a single voxel). Obviously, the authors can't test for all of these possibilities, but a more complete discussion of additional factors would strengthen the manuscript.

4. Did the pattern of V2/V3 topography depend at all on cortical surface depth? Even though the voxel size was relatively large, it should be possible to sample the voxel matrix using surfaces defined at different cortical depths relative to the gray matter. If the aberrant maps are somehow due to a distortion artifact associated with the cortical sampling and/or inflation procedure, the pattern may be less pronounced at deeper cortical depths.

5. Functional selectivity. In addition to pRF mapping, the HCP dataset included task runs, as well as resting state data. Did the pattern of activity in either task or resting state data follow the alternate boundaries for V2 and V3 defined here?

6. I wonder about morphometric/anatomical properties that may correlate or predict the alternative organization of V2/V3. Was there anything different in global brain structure (e.g., hemispheric asymmetry, brain size, etc) that might be predictive?

7. Did any demographic information predict the alternative pattern in V2/V3 (gender, age?)

8. Lastly, the results presented here suggest that the most widely used cortical atlases need to be revised to accommodate this known variability. It would be wonderful if the authors provided software to do this, or at least discuss the need for approaches that accommodate the range of variability described here.

*Reviewer #2 (Recommendations for the authors):*

The authors take aim at the retinotopic organization of V2 and V3, which in the field is generally assumed to follow a standard pattern: mirrored quarter-field representations abutting V1 on both the dorsal and ventral side. The deviations from the assumed canonical shape which the authors point to are intriguing, as this finding would change the way we think of the organization of low levels of the visual system. If we don't understand the lower levels of visual-cortical organization, what can we say about the higher levels? This finding could have quite a significant impact on how vision scientists approach their work.

Indeed, visually their point is well taken: the few visualisations in Figure 1 are quite compelling in their depiction of deviations of the locations and orientations of apparent visual field map boundaries.

I think the main point of the paper could be strengthened a lot, and that's because of the choices made in terms of the quantifications and analyses. In order for the field to incorporate the main finding into our way of thinking about low-level visual cortical organization, the findings need to be presented more convincingly. The manuscript fully depends on linear mixed effects modelling and clustering analysis based on summary results. The authors will want to dive more deeply into the phenomenology of what they're showing and be less concerned with statistical analyses.

One question that remains after reading the manuscript is what possible sources underpin the idiosyncratic polar angle patterns in so many participants.

For instance, I wonder whether the linear mixed effects modelling can adequately account for (i.e. correct for) the correlations of the polar angle deviations with mean EPI values and curvature. These effects, in my experience, definitely don't look like 'linear' effects in that they are quite all-or-none. Does a single 'omnibus' analysis suffice to prove that this pattern we see is a true signature of neural organization? I think many in the field, where drawing visual field map boundaries is a skill that's mastered over the course of many participants, would answer no. Similarly, the binarization of the patterns leading up to the clustering analysis could have a lot of unintended consequences. A more thorough visual exploration of these patterns would make the findings much more convincing. An open presentation of the data, so that the public can browse through the results, would be very valuable here.

Going back to the data:

1. To what extent do the patterns reported in the ms depend on their use of the 2mm HCP data? As the data were originally acquired at 1.6mm and these data can be readily downloaded, I wonder whether the idiosyncratic patterns in polar angle distribution on the surface depend on the pooling across voxels that is inherent in the data format that's being used here. If the authors re-fit those subjects with idiosyncratic polar angle patterns at higher resolution, does this change the pattern? It's likely that influences of venous signals are changed when subsampling the voxels.

2. In my experience, this sort of polar angle discontinuity can also be the result of anatomical segmentation errors, however small. I think the authors want to ensure that this is not a driving influence here.

3. To what extent could this sort of pattern of results be driven by eye movements? There are eye-tracking signals in the HCP dataset, so can the authors check this?

To get a bit of a better view of the authors' claims I quickly went through the surface maps provided by the NSD paper, since the authors cite this paper as an example study that also showcases the idiosyncrasies reported here. I'm uploading outlines of the most compelling deviations from the canonical V2/V3 organization in this dataset (See https://www.dropbox.com/s/auqjmiz1nghsxv3/Screenshot-Reviewer2.png?dl=0) To summarise; I see clear examples of the patterns outlined by the authors in 4 out of 16 hemispheres (admittedly, slightly conservative an estimate, perhaps). This is below what the authors report. For 2 or 3 of these instances, there is a clear venous eclipse-like effect that could have caused the polar angle patterns to deviate -- we know that this sort of thing is more likely to occur at 7T than at 3T, where retinotopic maps usually look smoother. I don't know what this means in relation to the reported findings, of course, but it could indicate that the more detailed preprocessing and higher single-subject quality of the NSD data decrease the occurrence of the reported patterns.

*Reviewer #3 (Recommendations for the authors):*

The work presented here by Ribeiro et al. attempts to quantify the observation that the "standard model" of V1, V2, and V3 matches the average human retinotopic maps but does not consistently match the retinotopic maps of individuals. (In this case, the "standard model" I'm referring to is just the assumption that the boundaries of V1, V2, and V3 topologically resemble those of the wedge-dipole model of Balasubramanian et al., 2002, or the banded double-sech model of Schira et al., 2010). Such non-standard retinotopic organizations in individual subjects are (as the authors note) well documented in other primates, and informal speculation about such non-standard human subjects has historically been common, if inconclusive, among those of us who study human V1-V3. Further, because the visual cortex is one of the few macroscopic brain systems for which neural computations, macroscopic organization, and connectivity are well-modeled, any novel observations about its individual variability are likely to lead to novel hypotheses about brain development and to serve as a template for future analyses. For these reasons, this manuscript is of clear value both to visual neuroscience researchers interested in the details of V1-V3 as well as to neuroscientists in adjacent domains.

While the findings of this paper are convincing and leave little doubt that non-standard organizations of V1-V3 not only exist but are common, it falls short of eliminating all possible explanations of such variability. To be clear, this is not primarily a critique of the paper but an observation that the authors' analyses point the field toward many additional questions. While the authors have done an excellent job demonstrating that large-scale deviations in the average curvature and BOLD signal do not explain large-scale variability of the maps, their analyses do not eliminate microscopic features of curvature or vasculature as potential causes. For example, a skeptic might claim that Cluster 2 (Figure 4) can be explained by partial voluming arising from tight curvature in a small patch of the dorsal V2-V3 region or by a blood vessel in the same region. If tight curvature or a vessel were critical to explaining these maps but were confined to a small patch of cortex, the analysis of curvature and BOLD intensity here may have been insufficiently precise to find it. (Of course, given the results of the analysis here, one might be justified in speculating that more detailed examinations of V1-V3 could reveal the "standard" model to be the model that most depend on typical sources of BOLD noise.) It would benefit the paper to include additional discussion of the extent to which the clear differences found in the clustered maps might or might not depend on microscopic features that are beyond the paper's current scope. However, the authors should be lauded for demonstrating conclusively that these differences cannot be easily dismissed.

I am particularly intrigued by the clustering analysis performed by the authors, summarized in Figure 4. This figure, along with Supplemental Figure 3, provides compelling evidence that the "Dorsal Y," as the branching lower vertical meridian representation of V1 and V2 is sometimes called, is common and reasonably consistent across individuals. However, I am not entirely convinced that clusters 1, 3, and 5 are capturing substantial deviations in the "standard" model of human retinotopy. It is possible that density-based or hierarchical clustering methods that do not employ any dimension reduction would yield different results (though I am skeptical that the differences would be especially large). Regardless, the fundamental issue with these three clusters is that the topology of the V1-V2-V3 boundaries does not appear terribly different, even when, for example, the intensity of the lower vertical meridian differs. Of course, the intensity of the representation of the meridian may be a feature of the maps worth examining in and of itself, but I suspect that most researchers will be more interested in the clear violation of the topology of visual area boundaries implicit in the "standard" model that one can easily see in Cluster 2. With respect to the differences between Cluster 2 and the other clusters, it seems likely to me that the authors could have captured these differences more easily and accurately by examining the distribution and cortical location of the visual field sign (see for example Tootell et al., 1998, DOI:10.1073/pnas.95.3.811). The most interesting feature of this cluster is arguably that it suggests either (1) that, in these subjects, V2 and V3 are split, with each having a "standard" portion as well as a dorsal-anterior representation; or alternately (2) that at least one (or more likely 2) additional visual areas exist just anterior to dorsal V2 and dorsal to anterior V1. My speculation here – that the patch of cortex bounded by the two branches of the Y (i.e., dorsal to anterior V1 and anterior to dorsal V2) must consist of 2 separate representations – arises from the fact that it contains two separate patches of cortex in which the field signs do not match.

Another possibility that I would have liked to see discussed is the likelihood that the "Dorsal Y" organization of Cluster 2 appears in many (more) subjects but does not appear at a consistent location with respect to eccentricity. To this end, I believe that Supplemental Figure 2 would have been more interesting had it been calculated not as a new set of clusters based on eccentricity but instead as the average eccentricity map of each of the clusters from Figure 4 that were based on the polar angle. My own observation (which I would like to emphasize is purely anecdotal) is that the eccentricity maps of subjects whose polar angle maps resemble those of Cluster 2 tend to bend sharply near the junction of the "y". I find myself wondering if the subjects in Cluster 2 have a similar feature in their eccentricity maps, whether the junction of the "y" of Cluster 2 tends to occur near the same eccentricity across subjects, and whether the cortical magnification functions of such subjects differ substantially from those who do not have this feature. The size of the PRFs in these regions might also be illuminating---do they scale with eccentricity in a manner that matches the rest of dorsal V2 and V3?

Overall, despite (and arguably because of) the many questions that this paper exposes, it is of substantial value to the community. Its results are straightforward and well-supported, and it provides a valuable starting point from which further refinements of the organization of the visual cortex can be studied.

Overall this paper is compelling and does not require substantial revision or refinement in order to be of great value to the field. To be frank, I have been hoping for several years that someone would publish a clear analysis of the dorsal retinotopic maps in V1-V3, and am pleased to see a conclusive demonstration of their variability finally appear. In my public review, I have pointed out many questions and implications that arise from these findings, and I believe that mentioning some of these implications would improve the paper. However, I do not believe that answering them is necessary for the paper – merely that discussion may be helpful for researchers who do not work on these maps closely on a regular basis.

One concrete suggestion I do have for the authors is that a cursory examination of the field sign may lead to clearer clusters and thus a clearer narrative of the data in Figure 4. That said, the field sign can be tricky – such an analysis would likely rest at least somewhat on light smoothing of the PRF parameters and may become difficult in ways I have not anticipated. Similarly, a brief analysis of the patch of cortex bounded by the arms of the "dorsal Y" in Cluster 2 would be of substantial interest and may help determine what kind of deviation from the traditional maps this cluster represents.

I would also suggest that the authors include some discussion of how the human clusters found in this analysis compare to the 6 clusters found in non-human primates. I don't think that much ink needs to be spent on this, but it struck me as odd that the non-human primate clusters would be cited as motivation for the clustering parameters in humans and then never discussed more carefully.

---

## [Author Response]

Essential revisions:1) Each of the reviewers raised questions about potential explanations for the alternate organization that the authors report. While it is likely not possible to completely address each of the reviewers' concerns with additional analysis, the authors should consider the reviewers' recommendations, and at the very least provide a serious discussion of the points that are not possible to address with further data analysis.

Here we briefly summarize our changes to the manuscript as suggested by the reviewers (each of these changes will be detailed throughout this letter):

Visualization – For seamless visualization of retinotopic maps as a function of the participant’s assignment to the reported clusters, we provide an executable code notebook with instructions for running them on Neurodesk (a data analysis environment for reproducible neuroimaging). In this code notebook, it is possible to interactively visualize maps and perform additional analyses of the retinotopic maps. Note that this code notebook and all source code are now available on GitHub (https://github.com/felenitaribeiro/VariabilityEarlyVisualCortex). In addition, we provide a tutorial for visualizing the 7T HCP Retinotopy dataset using Connectome Workbench on Neurodesk (https://www.neurodesk.org/tutorials-examples/tutorials/functional_imaging/connectomeworkbench/).

Detailed analyses for assessing the patch of cortex bounded by the arms of the "dorsal Y" – We provide extra analyses of the "dorsal Y" maps in the manuscript and make an interactive visualization of such analyses available in the same code notebook.

‘Limitations’ section – We now include a ‘Limitations’ subsection to expand our discussion about the limitations of our study, including the fact that the LME model does not eliminate microscopic features of the vasculature as a potential cause of individual variability in retinotopic maps.

‘Future directions’ section – Finally, we also included a ‘Future directions’ subsection to expand our discussion about many of the reviewers' suggestions that we could not address here or were beyond the scope of our study.

2) Both Reviewers 1 and 2 recommended providing a more systematic and complete visualization of the alternate organization that the authors report. There are a number of ways in which this could be accomplished. The authors should pursue an approach that most effectively documents the alternative pattern that they have discovered and the range of variability in the HCP data.

To address this concern, we have made available a Jupyter Notebook with interactive plotting functions and dropdown menus for visualizing an individual’s polar angle map (given their cluster assignment). In addition, we include five additional figure supplements (Figure 5—figure supplement 1-6) with nine randomly selected exemplary maps from the remaining clusters. Besides that, we provide a tutorial for visualizing the 7T HCP Retinotopy dataset using Connectome Workbench on Neurodesk (https://www.neurodesk.org/tutorials-examples/tutorials/functional_imaging/connectomeworkbench/).

Lines 822-833: “Data and code availability

The data used in this study is publicly available at BALSA (https://balsa.wustl.edu/study/show/9Zkk). In addition, all accompanying Python and MATLAB source codes are available on GitHub (https://github.com/felenitaribeiro/VariabilityEarlyVisualCortex). On our GitHub repository, executable code is available on interactive computational notebooks (Jupyter notebooks) that allow anyone to execute interactive plotting functions with dropdown menus to visualize an individual’s polar angle and visual field sign maps, given their cluster assignment. Note that our documentation provides instructions for running them on Neurodesk (Renton et al., 2022). Finally, the intermediate files for fitting the linear mixed effect models using Jamovi are available on the Open Science Framework (https://osf.io/tdkuj/).”

3) Reviewer 3 asked the authors to perform an examination of the field sign, which may shed light on the data in Figure 4., or perhaps a further analysis of the patch of cortex bounded by the arms of the "dorsal Y" in Cluster 2 would help elucidate the kind of deviation from the traditional maps this cluster represents.

As advised, we expanded our analysis of the “dorsal Y” in our manuscript by including polar angle gradient representation and the visual field sign analysis. Interestingly, the visual field sign analysis suggests that the area identified as dorsal V3 shows a discontinuity in the canonical mirror-image representation. This is now illustrated in Figure 7 (white circles added for emphasis). In addition, as previously mentioned, we made an interactive visualization of these analyses available in a code notebook. We hope to provide other visual neuroscientists a good starting point for exploring alternate models for early visual area parcellation.

Lines 800-821: “Visual field sign analysis

Lastly, we further examined unusual retinotopic maps to elucidate the kind of deviation from the canonical maps they represent. We performed a visual field sign analysis (Sereno et al., 1995, 1994), which combines both polar angle and eccentricity maps into a unique representation of the visual field as either a non-mirror-image (like V2) or a mirror-image representation of the retina (like V1). Since the left hemisphere represents the right visual field, which in polar angle includes 0⁰-90⁰ (upper right visual field) and 270⁰- 360⁰ (lower right visual field), we shifted the polar angle values so the point of wrap-around (from 360⁰ to 0⁰) was positioned at the horizontal meridian in the contralateral hemifield, avoiding the discontinuous representation between 360⁰ and 0⁰. Then, we interpolated the sparse and flattened polar angle and eccentricity maps onto a regular x-y grid using SciPy (Virtanen et al., 2020). Next, we determined the gradient of polar angle and eccentricity maps, mathematically expressed as:

Δfθ=∂fθ∂xi^+∂fθ∂yj^ and Δfr=∂fr∂xi^+∂fr∂yj^

where fθ is the polar angle map and fr the eccentricity, using the NumPy gradient numerical method (Harris et al., 2020). Finally, the angle between the polar angle and eccentricity maps’ gradient vectors was determined at each x-y coordinate. If the angle between the gradient vectors is between 0 and π, by convention, the cortical patch is a mirror-image representation of the retina; otherwise, it is a non-mirror-image. After binarizing the angle projection, we can conveniently infer borders between visual areas because adjacent areas often have the opposite visual field sign (Sereno et al., 1995; but see Yu et al. 2020 for caveat).”

Lines 291-340: “Visual field sign analysis for delineating visual area boundaries

Finally, we further examined retinotopic maps with a Y-shaped lower vertical representation, i.e., those primarily assigned to cluster 2, to elucidate the kind of deviation from the canonical maps they represent (Figure 7 and Figure 7—figure supplement 1). To do so, we performed a visual field sign analysis (Sereno et al., 1995, 1994), which combines both polar angle and eccentricity maps into a unique representation of the visual field as either a non-mirror-image or a mirror-image representation of the retina (Figure 7a; see Materials and methods). With such representation, we can directly infer visual area parcellation.

Figure 7b shows polar angle gradients in a ‘streamline’ representation and their respective visual field sign representation for two participants with canonical and four with Y-shaped lower vertical representations. While visual area boundaries in early visual cortex are conventionally identified by reversals in the progression of the polar angle values – or changes in the direction of polar angle gradients, it is unclear how to delineate boundaries in the dorsal portion of polar angle maps in those participants with non-canonical maps (note that their respective ventral portion followed the classical representation), but not on those with canonical maps. However, with the visual field sign representation, the boundaries delineating dorsal V2 in those participants with non-canonical maps are more explicit, and it reveals that the area identified as dorsal V3 shows a discontinuity in the expected mirror-image representation. Such representation has been proposed as the ‘incomplete-V3’ model of the third-tier cortex for the macaque (Angelucci and Rosa, 2015) and other similar models for the owl monkey (Sereno et al., 2015) and the marmoset monkey (Rosa and Tweedale, 2000). Figure 7—figure supplement 1 shows five more examples of polar angle maps with unusual Y-shaped lower vertical representations and five other examples with a truncated V3 boundary. While individuals with unusual Y-shaped lower vertical representations have a discontinuity in the canonical mirror-image representation of the retina in dorsal V3, individuals with a truncated V3 do not show such a discontinuity. Altogether, our findings may suggest that, at least in humans, the canonical model does not oppose other models established for non-human primates; these models coexist and reflect common modes of variability in the retinotopic mapping of early visual areas.”

4) There was a general consensus amongst the reviewers that the LME approach that the authors have employed may not be the most appropriate or the most sensitive analysis. For example, it would be good to demonstrate convincingly that including the normalized mean BOLD signal in the LME is in fact capable of detecting the venous eclipse around V4 (See Reviewer 1, point #2). Without such a positive result showing that the analysis can detect vascular artefacts, it is difficult to fully trust the lack of an effect in the early visual cortex.

We agree with the reviewers and would like to echo what Reviewer 3 mentioned, i.e., that the purpose of the LME model was to demonstrate that large-scale deviations in the average curvature and BOLD signal do not explain all the large-scale variability of the retinotopic maps. But, of course, this does not mean this method can eliminate microscopic (or local) features of curvature or vasculature as potential causes. As such, we reformulated our manuscript to make it more explicit. First, we provide more detail on how we computed pair-wise correlations among polar angle, eccentricity, curvature, and normalized mean BOLD signal for both V1-3 and each visual area separately (Figure 3; lines 206-214). Our findings reconcile with those in previous work. For example, we found that the polar angle representation is indeed correlated with the normalized mean BOLD signal, and the magnitude of such association varies across visual areas. Specifically, we found a higher correlation between polar angle and normalized BOLD signal maps in ventral visual areas than in dorsal areas. This difference could be explained by ventral areas' proximity to the dural venous sinuses (Winawer et al., 2010), i.e., the transverse sinus, the superior sagittal sinus, and the straight sinus. These venous sinuses are known to introduce artifacts to the BOLD signal, which might lead to changes in retinotopic maps. Importantly, none of these sinuses run near the dorsal V2/V3 (see Figure 4—figure supplement 2), and both of these dorsal areas show the lowest correlation (in magnitude) between the polar angle and normalized BOLD signal maps. But, again, it does not mean that vasculature does not affect retinotopy.

As for hV4, we found a weak correlation between the polar angle and normalized mean BOLD signal (LH: r = -0.06; RH: r = -0.07) and did not find an effect on *large-scale deviations* in an LME model. This finding does not reconcile with previous reports, i.e., that venous artifact impacts retinotopy. However, in Boyd Taylor et al. (2019), a fine-grained analysis of polar angle maps in hV4 indicated that “*[…] incomplete hV4 maps did not correspond with venous artefact in every instance, with incomplete maps being present in the absence of a venous eclipse and complete maps coexisting with a proximate venous eclipse.*” In our analysis, though, these results might reflect an inability to appropriately parcellate hV4 at the individual level when using an atlas-based parcellation; while early visual areas are more consistently found at specific spatial locations, this is not the case for other visual areas of which spatial location and extent seem to vary across participants. Unfortunately, we do not have manual segmentations of hV4 for the HCP dataset that would allow for a better understanding of the vasculature effect in retinotopic mapping. Accordingly, we extend our discussion about the limitations of our study in the manuscript.

Lines 711-719: “Accordingly, we calculated pair-wise correlations among polar angle, eccentricity, curvature, and normalized mean BOLD signal for both V1-3 and each visual area separately. Given a region of interest, each topographic map was vectorized, and data were concatenated across all participants (n=181) for each modality. Polar angle maps were converted such that 0⁰ corresponds to the horizontal meridian and 90⁰ corresponds to the upper and lower vertical meridians (Kurzawski et al., 2022). Finally, pair-wise correlations were determined using these concatenated sets of vectorized maps. Below we provide further motivation for considering each of these covariates.”

Lines 496-548: “Limitations

Although we demonstrate that common modes of deviation from the canonical dorsal V2 and V3 organization exist in the left hemisphere, further analyses are necessary to fully ascertain if such variability is indeed neurogenic or if it could be a result of measurement errors. Amongst potential sources of measurement error is the presence of large veins running adjacent to regions of interest. Accordingly, by using the normalized mean BOLD signal as a proxy for the location of large veins (Boyd Taylor et al., 2019; Kurzawski et al., 2022), studies have shown that voxels near these veins show lower mean BOLD signal, a phenomenon known as the venous eclipse, which affects pRF estimates, for example, in area hV4 (Boyd Taylor et al., 2019; Winawer et al., 2010). As such, deviations in the expected mean BOLD signal could result in deviations from the expected retinotopic organization. Thus, to better understand the potential effects of the presence of large veins on the different levels of variability in visual field representation across early visual areas, we considered both the pair-wise correlations between retinotopic maps and the normalized mean BOLD signal (Figure 3) and the large-scale deviation in the normalized mean BOLD signal as covariates in the LME model (Tables 3 and 4).

In the pair-wise correlation analysis, we found that the polar angle representation is indeed correlated with the normalized mean BOLD signal, and the magnitude of such association varies across visual areas. Specifically, we found a higher correlation (in magnitude) between the polar angle and normalized BOLD signal in ventral visual areas than in dorsal areas. This difference could be explained by ventral areas' proximity to the dural venous sinuses (Winawer et al., 2010), i.e., the transverse sinus, the superior sagittal sinus, and the straight sinus. These venous sinuses are known to introduce artifacts to the BOLD signal, which might lead to changes in retinotopic maps. Importantly, none of these sinuses run near the dorsal V2 and V3 (see Figure 4—figure supplement 2, in which it is possible to observe the likely confluence of sinuses), and both of which show the lowest correlation between polar angle and normalized BOLD signal maps.

Moreover, we modeled large-scale deviation in visual field maps using the large-scale deviation of the normalized mean BOLD signal as a covariate. We did find a significant effect of the normalized mean BOLD signal on the individual variability of eccentricity (Table 4) but not of polar angle maps (Table 3). Thus, to determine the effectiveness of the LME model for uncovering the effects of covariates, we similarly modeled the large-scale variability in polar angle maps of hV4 (not shown in the manuscript). We did not find an effect of the normalized mean BOLD signal on large-scale deviation in polar angle maps in hV4. However, we found a weak correlation between the polar angle and normalized mean BOLD signal (LH: r = -0.06; RH: r = -0.07). This finding does not reconcile with previous reports, i.e., that venous artifact impacts retinotopy. However, a fine-grained analysis of polar angle maps in hV4 indicated the inconsistent effect of the venous artifact on polar angle mapping (Boyd Taylor et al., 2019). In our analysis, though, these results might reflect an inability to appropriately parcellate hV4 at the individual level when using an atlas-based parcellation; while early visual areas are more consistently found at specific spatial locations, this is not the case for other visual areas of which spatial location and extent seem to vary across participants. Therefore, although we did not find a significant effect of the normalized mean BOLD signal in our LME model, it does not mean that the macro- and microvasculature do not affect retinotopy. For the former, future research might consider a fine-grained analysis of topographic deviations, such as the one reported by Boyd Taylor et al. (2019). For the latter, given that the mean BOLD signal is only used as a proxy for the location of large veins, a more detailed analysis of the microvasculature might require other imaging data, such as high-resolution time-of-flight magnetic resonance angiography data (Bollmann et al., 2022).”

Reviewer #1 (Recommendations for the authors):This manuscript concerns the organization of the first three cortical visual areas (V1/V2/V3). For decades, the field has assumed an organization in which V2 and V3 are conceived as alternating, mirror-symmetric quarterfield representations emanating from a full hemifield field representation in V1. Here, the authors have described an innovative and (mostly) compelling analysis demonstrating a striking degree of intersubject variability in this organization. Deviations from the canonical organization are not random. Instead, the authors have uncovered a subset of subjects that follow an alternative organization, most prevalent in the dorsal portion of the left hemisphere.This study comes as a breath of fresh air, a huge relief to somebody who has personally spent many hundreds of hours over the years hand-drawing visual area boundaries on inflated brains. As anybody who has attempted this task can readily attest, there is indeed a large degree of inter-subject variability, and forcing the canonical organization on some hemispheres can involve more than a bit of guesswork. The present study suggests that the glove simply doesn't fit for these subjects.I have always assumed that the canonical organization is correct and that the frequent deviations that I've observed were due to measurement errors. My biggest criticism of the present study is that the authors have not gone far enough in demonstrating that this isn't the case. In other words, I am not completely convinced that this alternative organization derives from variability in the neural map, rather than from a systematic artifact, either in the MRI measurements (functional or structural) or in one of the many processing stages to intervene between data acquisition and the final map.Regardless of whether the results shown here are indeed neurogenic, or are somehow the result of measurement error, the implications are far-reaching. Over the years, countless fMRI studies have shown bar graphs quantifying some effect in areas V1/V2/V3, with areas boundaries that were either drawn by hand or estimated with an atlas, conforming to the standard, canonical organization. A major implication of the current results is that these previous studies may have mischaracterized differences between these visual areas. And if that is the case in the early visual cortex, I am sure this is a much larger problem in higher-order cortical areas. So in sum, I think this is a really important set of observations and I am confident that it will have an impact on the field.1. The manuscript is very clear and well-written and I am inclined to believe the results. But this inclination is really based on simple visual inspection of the maps shown, and not the statistical analyses per se. Given that the I find visual inspection to be the most compelling presentation of the result, the manuscript could be greatly improved by showing more data visually demonstrating the range of maps in the HCP data. This could be a supplement, or a pointer to an online resource showing the retinotopic maps for all subjects and all hemispheres, perhaps categorized into meaningful groups.

To address this concern, we have made available a Jupyter Notebook with interactive plotting functions and dropdown menus for visualizing an individual’s polar angle map (given their cluster assignment) on GitHub (https://github.com/felenitaribeiro/VariabilityEarlyVisualCortex). In addition, we include five additional figure supplements (Figure 5—figure supplement 1-6) with nine randomly selected exemplary maps from the remaining clusters.

Lines 822-833: “Data and code availability

The data used in this study is publicly available at BALSA (https://balsa.wustl.edu/study/show/9Zkk). In addition, all accompanying Python and MATLAB source codes are available on GitHub (https://github.com/felenitaribeiro/VariabilityEarlyVisualCortex). On our GitHub repository, executable code is available on interactive computational notebooks (Jupyter notebooks) that allow anyone to execute interactive plotting functions with dropdown menus to visualize an individual’s polar angle and visual field sign maps, given their cluster assignment. Note that our documentation provides instructions for running them on Neurodesk (Renton et al., 2022). Finally, the intermediate files for fitting the linear mixed effect models using Jamovi are available on the Open Science Framework (https://osf.io/tdkuj/).”

2. The authors have not done an especially convincing job of discounting various artifacts that may have contributed to the variability in map structure. The authors included covariates in the LME for curvature, normalized mean BOLD signal, and intra-individual variability in pRF estimates, and then argue that these covariates do not account for the pattern of variability in map organization. But doesn't this logic ask us to accept the null hypothesis that these factors aren't playing a role? "line 367: Our findings indicate that the main effects found here were not a mere reflection of variation in the reliability of the individual maps and other covariates." But in fact, the analysis simply suggests that an effect could not be detected. It would be more convincing if the authors could demonstrate that such effects do occur and do impact retinotopic maps (such as the venous eclipse in V4), but that they aren't playing a role here. Specifically, I am concerned that including the normalized mean BOLD signal in an LME isn't a very sensitive way of testing for the effects of the vasculature. Demonstrating a positive effect in V4 with this approach (where we know all about the impact of a large draining vein) would be a more convincing demonstration of the method, and enable us to interpret the lack of an effect in V2/V3.

We have considered this suggestion carefully. As a result, to determine the effectiveness of the LME model for uncovering the effects of covariates, we similarly modeled the large-scale variability in polar angle maps of hV4 as a function of hemispheres and included the individual variability in normalized mean BOLD signal as a covariate. We did not find an effect of normalized mean BOLD signal on large-scale deviation in polar angle maps. However, we found a weak correlation between the polar angle and normalized mean BOLD signal (LH: r = -0.06; RH: r = -0.07). This finding does not entirely reconcile with previous reports that venous artifact impacts on the assessment of retinotopy in hV4. However, our results might reflect an inability to appropriately parcellate hV4 at the individual level when using an atlas-based parcellation. We now discuss this important limitation in our manuscript and have also emphasized throughout the manuscript that such an analysis cannot rule out the effect of vasculature in retinotopy, as the reviewer has mentioned.

Lines 496-548: “Limitations

Although we demonstrate that common modes of deviation from the canonical dorsal V2 and V3 organization exist in the left hemisphere, further analyses are necessary to fully ascertain if such variability is indeed neurogenic or if it could be a result of measurement errors. Amongst potential sources of measurement error is the presence of large veins running adjacent to regions of interest. Accordingly, by using the normalized mean BOLD signal as a proxy for the location of large veins (Boyd Taylor et al., 2019; Kurzawski et al., 2022), studies have shown that voxels near these veins show lower mean BOLD signal, a phenomenon known as the venous eclipse, which affects pRF estimates, for example, in area hV4 (Boyd Taylor et al., 2019; Winawer et al., 2010). As such, deviations in the expected mean BOLD signal could result in deviations from the expected retinotopic organization. Thus, to better understand the potential effects of the presence of large veins on the different levels of variability in visual field representation across early visual areas, we considered both the pair-wise correlations between retinotopic maps and the normalized mean BOLD signal (Figure 3) and the large-scale deviation in the normalized mean BOLD signal as covariates in the LME model (Tables 3 and 4). In the pair-wise correlation analysis, we found that the polar angle representation is indeed correlated with the normalized mean BOLD signal, and the magnitude of such association varies across visual areas. Specifically, we found a higher correlation (in magnitude) between the polar angle and normalized BOLD signal in ventral visual areas than in dorsal areas. This difference could be explained by ventral areas' proximity to the dural venous sinuses (Winawer et al., 2010), i.e., the transverse sinus, the superior sagittal sinus, and the straight sinus. These venous sinuses are known to introduce artifacts to the BOLD signal, which might lead to changes in retinotopic maps. Importantly, none of these sinuses run near the dorsal V2 and V3 (see Figure 4—figure supplement 2, in which it is possible to observe the likely confluence of sinuses), and both of which show the lowest correlation between polar angle and normalized BOLD signal maps. Moreover, we modeled large-scale deviation in visual field maps using the large-scale deviation of the normalized mean BOLD signal as a covariate. We did find a significant effect of the normalized mean BOLD signal on the individual variability of eccentricity (Table 4) but not of polar angle maps (Table 3). Thus, to determine the effectiveness of the LME model for uncovering the effects of covariates, we similarly modeled the large-scale variability in polar angle maps of hV4 (not shown in the manuscript). We did not find an effect of the normalized mean BOLD signal on large-scale deviation in polar angle maps in hV4. However, we found a weak correlation between the polar angle and normalized mean BOLD signal (LH: r = -0.06; RH: r = -0.07). This finding does not reconcile with previous reports, i.e., that venous artifact impacts retinotopy. However, a fine-grained analysis of polar angle maps in hV4 indicated the inconsistent effect of the venous artifact on polar angle mapping (Boyd Taylor et al., 2019).

In our analysis, though, these results might reflect an inability to appropriately parcellate hV4 at the individual level when using an atlas-based parcellation; while early visual areas are more consistently found at specific spatial locations, this is not the case for other visual areas of which spatial location and extent seem to vary across participants. Therefore, although we did not find a significant effect of the normalized mean BOLD signal in our LME model, it does not mean that the macro- and microvasculature do not affect retinotopy. For the former, future research might consider a fine-grained analysis of topographic deviations, such as the one reported by Boyd Taylor et al. (2019). For the latter, given that the mean BOLD signal is only used as a proxy for the location of large veins, a more detailed analysis of the microvasculature might require other imaging data, such as high-resolution time-of-flight magnetic resonance angiography data (Bollmann et al., 2022).”

3. In addition to the three covariates that the authors considered, I wonder about additional issues, such as segmentation reliability, cortical thickness, partial voluming, and bleed over from the wall of the adjacent sulcal surface (i.e., multiple surfaces passing through a single voxel). Obviously, the authors can't test for all of these possibilities, but a more complete discussion of additional factors would strengthen the manuscript.

As suggested, we have expanded our discussion regarding the abovementioned confounders.

Lines 453-476: “Retinotopic maps could also vary as a function of data resolution and cortical depth. For example, signal-to-noise ratio (SNR) and partial volume artifacts are directly affected by data resolution (or voxel size); that is, both reduce with voxel size (Hoffmann et al., 2009). While lower SNR might lead to noisier (or less smooth) retinotopic maps (Hoffmann et al., 2009), the gain from the reduced susceptibility to partial volume artifacts will likely result in increased validity of the observed maps. Partial volume artifacts may arise from patches of opposite walls of a sulcus running across a single voxel or even small vessels, leading to inaccurate signals from the combination of different brain regions and tissues. With increasing magnetic field strength, it should be easier to strike the right balance between high SNR and low partial volume artifacts, which is crucial for determining the impact of registration errors due to partial volume artifacts on the variability of retinotopic maps. Moreover, previous studies have also shown that the hemodynamic response function (Puckett et al., 2016) and a spatial pattern of activation (Polimeni et al., 2010) varied across depths in V1. Specifically, Polimeni and colleagues found that a spatial pattern of activation (an ‘M’) becomes clearer from the white matter surface to the mid-thickness surfaces and then deteriorates once again near the pial surface. Altogether, these studies motivate a more thorough investigation of how retinotopy, as measured by fMRI, varies as a function of data resolution and cortical depth and its implication on individual variability in retinotopy. However, it is also important to note that studies of the columnar organization of non-human primates using single-cell recordings have not found any evidence that the receptive field location varies with cortical depth, although the receptive field size changes, being smallest in the middle layers (Hubel and Wiesel, 1974; Rosa et al., 1997).”

4. Did the pattern of V2/V3 topography depend at all on cortical surface depth? Even though the voxel size was relatively large, it should be possible to sample the voxel matrix using surfaces defined at different cortical depths relative to the gray matter. If the aberrant maps are somehow due to a distortion artifact associated with the cortical sampling and/or inflation procedure, the pattern may be less pronounced at deeper cortical depths.

We have yet to investigate whether V2/V3 topography varies according to cortical depth. With the current data (voxel size is equal to 1.6 mm isotropic), it might be difficult to reliably tell whether these patterns vary according to layers, given that these cortical areas are thin (2-2.5mm). However, with higher resolution data, previous studies have shown that the hemodynamic response function (0.8 mm isotropic; Puckett et al., 2016) and a spatial pattern of activation (1 mm isotropic; Polimeni et al., 2010) varied across depths in V1. Specifically, Polimeni and colleagues found that the spatial pattern of activation (an ‘M’) becomes clearer from the white matter surface to the mid-thickness surfaces and then deteriorates once again near the pial surface. Here, we limited our analysis to pRF fits promptly available at BALSA (https://balsa.wustl.edu/study/show/9Zkk). These data have gone through the HCP pre-processing pipelines (Glasser et al., 2013; Benson et al., 2018), of which only data sampled to the mid-thickness surface was completely pre-processed and used for pRF mapping (Benson et al., 2018).

Lines 453-476: “Retinotopic maps could also vary as a function of data resolution and cortical depth. For example, signal-to-noise ratio (SNR) and partial volume artifacts are directly affected by data resolution (or voxel size); that is, both reduce with voxel size (Hoffmann et al., 2009). While lower SNR might lead to noisier (or less smooth) retinotopic maps (Hoffmann et al., 2009), the gain from the reduced susceptibility to partial volume artifacts will likely result in increased validity of the observed maps. Partial volume artifacts may arise from patches of opposite walls of a sulcus running across a single voxel or even small vessels, leading to inaccurate signals from the combination of different brain regions and tissues. With increasing magnetic field strength, it should be easier to strike the right balance between high SNR and low partial volume artifacts, which is crucial for determining the impact of registration errors due to partial volume artifacts on the variability of retinotopic maps. Moreover, previous studies have also shown that the hemodynamic response function (Puckett et al., 2016) and a spatial pattern of activation (Polimeni et al., 2010) varied across depths in V1. Specifically, Polimeni and colleagues found that a spatial pattern of activation (an ‘M’) becomes clearer from the white matter surface to the mid-thickness surfaces and then deteriorates once again near the pial surface. Altogether, these studies motivate a more thorough investigation of how retinotopy, as measured by fMRI, varies as a function of data resolution and cortical depth and its implication on individual variability in retinotopy. However, it is also important to note that studies of the columnar organization of non-human primates using single-cell recordings have not found any evidence that the receptive field location varies with cortical depth, although the receptive field size changes, being smallest in the middle layers (Hubel and Wiesel, 1974; Rosa et al., 1997).”

5. Functional selectivity. In addition to pRF mapping, the HCP dataset included task runs, as well as resting state data. Did the pattern of activity in either task or resting state data follow the alternate boundaries for V2 and V3 defined here?

We did not investigate differences in functional selectivity in V1-V3. The battery of tasks in the HCP dataset includes working memory, gambling, motor, language, social cognition, and relational and emotional processing. These tasks evaluate cognitive processes and might not be the most appropriate for investigating functional selectivity in early visual areas. However, such an analysis would be valuable since misidentifying individual visual areas can lead to the misinterpretation of functional selectivity findings. Hence, we have expanded this discussion in Future Directions.

Lines 550-568: “Our findings raise questions about if and how cortical atlases should be revisited to accommodate deviations from the canonical model of retinotopic organization, especially for dorsal V3. Here, by using the visual field sign representation, we could better understand what kind of deviation from the canonical model atypical maps represent. Therefore, using such data representation could be helpful for the manual segmentation of atypical maps. Alternatively, combining deep learning models (Ribeiro et al., 2021) for generating a retinotopic prior that accommodates more variability with a Bayesian framework (Benson and Winawer, 2018) for boundary delineation might prove fruitful to support the need for automated individual-level parcellation methods. It would also be desirable to use functional characteristics of areas, such as the responses to specific types of visual stimulation, to increase the confidence in assignment of boundaries. Whereas this may be possible in some situations (e.g. using motion selectivity as a localizer for area MT; Pitzalis et al., 2010), attempts to segregate V3 from adjacent areas on this basis may be more challenging, due to the physiological similarity between this area and the adjacent V2 and V3a (Gegenfurtner et al., 1997; Levitt et al., 1994; Zeki, 1978). Differences in pRF size (Zhu and Vanduffel, 2019) could offer some insight, although the wide overlap in the distributions of single-unit receptive field sizes in adjacent areas (Rosa, 1997) suggests that obtaining clear-cut boundaries on this basis remains unlikely.”

6. I wonder about morphometric/anatomical properties that may correlate or predict the alternative organization of V2/V3. Was there anything different in global brain structure (e.g., hemispheric asymmetry, brain size, etc) that might be predictive?7. Did any demographic information predict the alternative pattern in V2/V3 (gender, age?)

To determine whether other properties would predict individual variability in polar angle maps, we included additional covariates in our LME model. Specifically, this model is similar to the one reported in Table 3, but it also has age, gender (as cofactor), and gray matter volume as additional covariates. In brief, we did not find significant effects of any of these variables (https://osf.io/c2pjm).

Lines 202-206: “To determine whether other properties would predict individual variability in polar angle maps, we included age, gender (as cofactor), and gray matter volume as additional covariates, but did not find significant effects of any of these variables (not shown here, but available as Supplementary File 5).”

8. Lastly, the results presented here suggest that the most widely used cortical atlases need to be revised to accommodate this known variability. It would be wonderful if the authors provided software to do this, or at least discuss the need for approaches that accommodate the range of variability described here.

As suggested, we have expanded our discussion about this need in our manuscript.

Lines 550-594: “Our findings raise questions about if and how cortical atlases should be revisited to accommodate deviations from the canonical model of retinotopic organization, especially for dorsal V3. Here, by using the visual field sign representation, we could better understand what kind of deviation from the canonical model atypical maps represent. Therefore, using such data representation could be helpful for the manual segmentation of atypical maps. Alternatively, combining deep learning models (Ribeiro et al., 2021) for generating a retinotopic prior that accommodates more variability with a Bayesian framework (Benson and Winawer, 2018) for boundary delineation might prove fruitful to support the need for automated individual-level parcellation methods. It would also be desirable to use functional characteristics of areas, such as the responses to specific types of visual stimulation, to increase the confidence in assignment of boundaries. Whereas this may be possible in some situations (e.g. using motion selectivity as a localizer for area MT; Pitzalis et al., 2010), attempts to segregate V3 from adjacent areas on this basis may be more challenging, due to the physiological similarity between this area and the adjacent V2 and V3a (Gegenfurtner et al., 1997; Levitt et al., 1994; Zeki, 1978). Differences in pRF size (Zhu and Vanduffel, 2019) could offer some insight, although the wide overlap in the distributions of single-unit receptive field sizes in adjacent areas (Rosa, 1997) suggests that obtaining clear-cut boundaries on this basis remains unlikely.

Importantly, accommodating individual variability for the automatic parcellation of visual areas is also crucial for understanding the functional properties of the human visual cortex. These analyses require the precise delineation of boundaries between visual areas, either by manually tracing transitions in visual field maps or by using an automatic segmentation method (Benson et al., 2014; Benson and Winawer, 2018; Dougherty et al., 2003). In both cases, a spatially consistent mapping (i.e., a canonical representation) of continuous, alternating bands of vertical and horizontal meridian representation in V2 and V3 is often assumed. However, we demonstrate that deviations from the canonical model exist, especially in the left hemisphere and dorsal V3. This finding could have several implications for post hoc analyses requiring visual area delineation, suggesting that previous studies may have mischaracterized differences between these visual areas due to misidentification. Therefore, an important future direction of this work is determining how functional selectivity varies across visual areas parcellated according to the canonical model compared to using new parcels estimated with the visual field sign analysis.

In addition, the present findings highlight the need for a more comprehensive assessment of the degree of variability in visuotopic maps in non-human primates, where a higher degree of precision can be achieved with invasive methods including single neuron recordings and optical imaging of intrinsic signals. To date, variability has only been reported in macaque monkeys (Gattass et al. 1988), but the available data in marmoset and owl monkeys indicate a reproducible organization that does not fully agree with the canonical model of dorsal V3 (Angelucci and Rosa, 2015; Rosa and Schmid, 1995; Rosa and Tweedale, 2000; Sereno et al., 2015). Whether this simply reflects the small number of individuals explored, or a truly more stable configuration (perhaps associated with the larger brain size in humans; Angelucci and Rosa, 2015; Rosa and Tweedale, 2005) remains to be determined.”

Reviewer #2 (Recommendations for the authors):The authors take aim at the retinotopic organization of V2 and V3, which in the field is generally assumed to follow a standard pattern: mirrored quarter-field representations abutting V1 on both the dorsal and ventral side. The deviations from the assumed canonical shape which the authors point to are intriguing, as this finding would change the way we think of the organization of low levels of the visual system. If we don't understand the lower levels of visual-cortical organization, what can we say about the higher levels? This finding could have quite a significant impact on how vision scientists approach their work.Indeed, visually their point is well taken: the few visualisations in Figure 1 are quite compelling in their depiction of deviations of the locations and orientations of apparent visual field map boundaries.I think the main point of the paper could be strengthened a lot, and that's because of the choices made in terms of the quantifications and analyses. In order for the field to incorporate the main finding into our way of thinking about low-level visual cortical organization, the findings need to be presented more convincingly. The manuscript fully depends on linear mixed effects modelling and clustering analysis based on summary results. The authors will want to dive more deeply into the phenomenology of what they're showing and be less concerned with statistical analyses.

Aiming to elucidate the deviation from traditional maps that some of the atypical retinotopic maps represent, we now provide additional analyses of the "dorsal Y" maps in the manuscript. We include the representation of polar angle maps’ gradients and their corresponding visual field sign representations. In brief, the visual field sign analysis suggests that the region of cortex usually identified as dorsal V3 shows a discontinuity in the canonical mirror-image representation of the retina in many individuals.

Lines 800-821: “Visual field sign analysis

Lastly, we further examined unusual retinotopic maps to elucidate the kind of deviation from the canonical maps they represent. We performed a visual field sign analysis (Sereno et al., 1995, 1994), which combines both polar angle and eccentricity maps into a unique representation of the visual field as either a non-mirror-image (like V2) or a mirror-image representation of the retina (like V1). Since the left hemisphere represents the right visual field, which in polar angle includes 0⁰-90⁰ (upper right visual field) and 270⁰- 360⁰ (lower right visual field), we shifted the polar angle values so the point of wrap-around (from 360⁰ to 0⁰) was positioned at the horizontal meridian in the contralateral hemifield, avoiding the discontinuous representation between 360⁰ and 0⁰. Then, we interpolated the sparse and flattened polar angle and eccentricity maps onto a regular x-y grid using SciPy (Virtanen et al., 2020). Next, we determined the gradient of polar angle and eccentricity maps, mathematically expressed as:

Δfθ=∂fθ∂xi^+∂fθ∂yj^ and Δfr=∂fr∂xi^+∂fr∂yj^

where fθ is the polar angle map and fr the eccentricity, using the NumPy gradient numerical method (Harris et al., 2020). Finally, the angle between the polar angle and eccentricity maps’ gradient vectors was determined at each x-y coordinate. If the angle between the gradient vectors is between 0 and π, by convention, the cortical patch is a mirror-image representation of the retina; otherwise, it is a non-mirror-image. After binarizing the angle projection, we can conveniently infer borders between visual areas because adjacent areas often have the opposite visual field sign (Sereno et al., 1995; but see Yu et al. 2020 for caveat).”

Lines 291-340: “Visual field sign analysis for delineating visual area boundaries

Finally, we further examined retinotopic maps with a Y-shaped lower vertical representation, i.e., those primarily assigned to cluster 2, to elucidate the kind of deviation from the canonical maps they represent (Figure 7 and Figure 7-figurse supplement 2). To do so, we performed a visual field sign analysis (Sereno et al., 1995, 1994), which combines both polar angle and eccentricity maps into a unique representation of the visual field as either a non-mirror-image or a mirror-image representation of the retina (Figure 7a; see Materials and methods). With such representation, we can directly infer visual area parcellation.

Figure 7b shows polar angle gradients in a ‘streamline’ representation and their respective visual field sign representation for two participants with canonical and four with Y-shaped lower vertical representations. While visual area boundaries in early visual cortex are conventionally identified by reversals in the progression of the polar angle values – or changes in the direction of polar angle gradients, it is unclear how to delineate boundaries in the dorsal portion of polar angle maps in those participants with non-canonical maps (note that their respective ventral portion followed the classical representation), but not on those with canonical maps. However, with the visual field sign representation, the boundaries delineating dorsal V2 in those participants with non-canonical maps are more explicit, and it reveals that the area identified as dorsal V3 shows a discontinuity in the expected mirror-image representation. Such representation has been proposed as the ‘incomplete-V3’ model of the third-tier cortex for the macaque (Angelucci and Rosa, 2015) and other similar models for the owl monkey (Sereno et al., 2015) and the marmoset monkey (Rosa and Tweedale, 2000). Figure 7—figure supplement 1 shows five more examples of polar angle maps with unusual Y-shaped lower vertical representations and five other examples with a truncated V3 boundary. While individuals with unusual Y-shaped lower vertical representations have a discontinuity in the canonical mirror-image representation of the retina in dorsal V3, individuals with a truncated V3 do not show such a discontinuity. Altogether, our findings may suggest that, at least in humans, the canonical model does not oppose other models established for non-human primates; these models coexist and reflect common modes of variability in the retinotopic mapping of early visual areas.”

One question that remains after reading the manuscript is what possible sources underpin the idiosyncratic polar angle patterns in so many participants.For instance, I wonder whether the linear mixed effects modelling can adequately account for (i.e. correct for) the correlations of the polar angle deviations with mean EPI values and curvature. These effects, in my experience, definitely don't look like 'linear' effects in that they are quite all-or-none. Does a single 'omnibus' analysis suffice to prove that this pattern we see is a true signature of neural organization? I think many in the field, where drawing visual field map boundaries is a skill that's mastered over the course of many participants, would answer no. Similarly, the binarization of the patterns leading up to the clustering analysis could have a lot of unintended consequences. A more thorough visual exploration of these patterns would make the findings much more convincing. An open presentation of the data, so that the public can browse through the results, would be very valuable here.

For a more thorough visual exploration of the retinotopic maps and clusters, we have made a Jupyter Notebook available with interactive plotting functions and dropdown menus for visualizing an individual’s polar angle map (given their cluster assignment). In addition, we include five additional figure supplements (Figure 5—figure supplement 1-6) with nine randomly selected exemplary maps from the remaining clusters. Besides that, we provide a tutorial for visualizing the 7T HCP Retinotopy dataset using Connectome Workbench on Neurodesk (https://www.neurodesk.org/tutorials-examples/tutorials/functional_imaging/connectomeworkbench/).

As for the concern about the appropriateness of the linear mixed effects model to account for the covariates discussed in the manuscript, we similarly modeled the large-scale variability in polar angle maps of hV4 as a function of hemispheres and included the individual variability in normalized mean BOLD signal as a covariate. We did not find an effect of normalized mean BOLD signal on large-scale deviation in polar angle maps. However, we found a weak correlation between the polar angle and normalized mean BOLD signal (LH: r = -0.06; RH: r = -0.07). This finding does not reconcile with previous reports, i.e., that venous artifact impacts retinotopy. Our results might reflect an inability to appropriately parcellate hV4 at the individual level when using an atlas-based parcellation. We now discuss this important limitation in our manuscript and have also emphasized throughout the manuscript that such an analysis cannot rule out the effect of vasculature in retinotopy.

Lines 496-548: “Limitations

Although we demonstrate that common modes of deviation from the canonical dorsal V2 and V3 organization exist in the left hemisphere, further analyses are necessary to fully ascertain if such variability is indeed neurogenic or if it could be a result of measurement errors. Amongst potential sources of measurement error is the presence of large veins running adjacent to regions of interest. Accordingly, by using the normalized mean BOLD signal as a proxy for the location of large veins (Boyd Taylor et al., 2019; Kurzawski et al., 2022), studies have shown that voxels near these veins show lower mean BOLD signal, a phenomenon known as the venous eclipse, which affects pRF estimates, for example, in area hV4 (Boyd Taylor et al., 2019; Winawer et al., 2010). As such, deviations in the expected mean BOLD signal could result in deviations from the expected retinotopic organization. Thus, to better understand the potential effects of the presence of large veins on the different levels of variability in visual field representation across early visual areas, we considered both the pair-wise correlations between retinotopic maps and the normalized mean BOLD signal (Figure 3) and the large-scale deviation in the normalized mean BOLD signal as covariates in the LME model (Tables 3 and 4).

In the pair-wise correlation analysis, we found that the polar angle representation is indeed correlated with the normalized mean BOLD signal, and the magnitude of such association varies across visual areas. Specifically, we found a higher correlation (in magnitude) between the polar angle and normalized BOLD signal in ventral visual areas than in dorsal areas. This difference could be explained by ventral areas' proximity to the dural venous sinuses (Winawer et al., 2010), i.e., the transverse sinus, the superior sagittal sinus, and the straight sinus. These venous sinuses are known to introduce artifacts to the BOLD signal, which might lead to changes in retinotopic maps. Importantly, none of these sinuses run near the dorsal V2 and V3 (see Figure 4—figure supplement 2, in which it is possible to observe the likely confluence of sinuses), and both of which show the lowest correlation between polar angle and normalized BOLD signal maps.

Moreover, we modeled large-scale deviation in visual field maps using the large-scale deviation of the normalized mean BOLD signal as a covariate. We did find a significant effect of the normalized mean BOLD signal on the individual variability of eccentricity (Table 4) but not of polar angle maps (Table 3). Thus, to determine the effectiveness of the LME model for uncovering the effects of covariates, we similarly modeled the large-scale variability in polar angle maps of hV4 (not shown in the manuscript). We did not find an effect of the normalized mean BOLD signal on large-scale deviation in polar angle maps in hV4. However, we found a weak correlation between the polar angle and normalized mean BOLD signal (LH: r = -0.06; RH: r = -0.07). This finding does not reconcile with previous reports, i.e., that venous artifact impacts retinotopy. However, a fine-grained analysis of polar angle maps in hV4 indicated the inconsistent effect of the venous artifact on polar angle mapping (Boyd Taylor et al., 2019). In our analysis, though, these results might reflect an inability to appropriately parcellate hV4 at the individual level when using an atlas-based parcellation; while early visual areas are more consistently found at specific spatial locations, this is not the case for other visual areas of which spatial location and extent seem to vary across participants. Therefore, although we did not find a significant effect of the normalized mean BOLD signal in our LME model, it does not mean that the macro- and microvasculature do not affect retinotopy. For the former, future research might consider a fine-grained analysis of topographic deviations, such as the one reported by Boyd Taylor et al. (2019). For the latter, given that the mean BOLD signal is only used as a proxy for the location of large veins, a more detailed analysis of the microvasculature might require other imaging data, such as high-resolution time-of-flight magnetic resonance angiography data (Bollmann et al., 2022).”

Going back to the data:1. To what extent do the patterns reported in the ms depend on their use of the 2mm HCP data? As the data were originally acquired at 1.6mm and these data can be readily downloaded, I wonder whether the idiosyncratic patterns in polar angle distribution on the surface depend on the pooling across voxels that is inherent in the data format that's being used here. If the authors re-fit those subjects with idiosyncratic polar angle patterns at higher resolution, does this change the pattern? It's likely that influences of venous signals are changed when subsampling the voxels.

Here, we limited our analysis to pRF fits promptly available at BALSA (https://balsa.wustl.edu/study/show/9Zkk), which have gone through the HCP pre-processing pipelines (Glasser et al., 2013) and only include three pRF fits for the mid-thickness surface at 2 mm spatial resolution (Benson et al., 2018). However, given that the signal-to-noise ratio (SNR) decreases with voxel size (Hoffman et al., 2009), we would expect that, for a fixed variance explained threshold, the functional data sampled at a higher resolution would result in noisier (or less smooth) retinotopic maps. We expanded our discussion about these concerns in the manuscript.

Lines 453-476: “Retinotopic maps could also vary as a function of data resolution and cortical depth. For example, signal-to-noise ratio (SNR) and partial volume artifacts are directly affected by data resolution (or voxel size); that is, both reduce with voxel size (Hoffmann et al., 2009). While lower SNR might lead to noisier (or less smooth) retinotopic maps (Hoffmann et al., 2009), the gain from the reduced susceptibility to partial volume artifacts will likely result in increased validity of the observed maps. Partial volume artifacts may arise from patches of opposite walls of a sulcus running across a single voxel or even small vessels, leading to inaccurate signals from the combination of different brain regions and tissues. With increasing magnetic field strength, it should be easier to strike the right balance between high SNR and low partial volume artifacts, which is crucial for determining the impact of registration errors due to partial volume artifacts on the variability of retinotopic maps. Moreover, previous studies have also shown that the hemodynamic response function (Puckett et al., 2016) and a spatial pattern of activation (Polimeni et al., 2010) varied across depths in V1. Specifically, Polimeni and colleagues found that a spatial pattern of activation (an ‘M’) becomes clearer from the white matter surface to the mid-thickness surfaces and then deteriorates once again near the pial surface. Altogether, these studies motivate a more thorough investigation of how retinotopy, as measured by fMRI, varies as a function of data resolution and cortical depth and its implication on individual variability in retinotopy. However, it is also important to note that studies of the columnar organization of non-human primates using single-cell recordings have not found any evidence that the receptive field location varies with cortical depth, although the receptive field size changes, being smallest in the middle layers (Hubel and Wiesel, 1974; Rosa et al., 1997).”

2. In my experience, this sort of polar angle discontinuity can also be the result of anatomical segmentation errors, however small. I think the authors want to ensure that this is not a driving influence here.

The three pRF model fits and other pre-processed data analyzed and reported in the manuscript (Supplementary File 7) have gone through the HCP pre-processing pipeline, which includes PreFreeSurfer, FreeSurfer, and PostFreeSurfer pipelines (Benson et al., 2018; Glasser et al., 2013). Although it could be the case that ‘one pre-processing pipeline does not fit all,’ we believe that given the complexity of the HCP pipeline, such an investigation is beyond the scope of our manuscript.

3. To what extent could this sort of pattern of results be driven by eye movements? There are eye-tracking signals in the HCP dataset, so can the authors check this?

We checked the eye-tracking signal as suggested by the reviewer and did not find a consistent change in gaze position in individuals across clusters. Note that, as reported by Benson et al. (2018), “[…] eye tracking data are available on ConnectomeDB for most subjects, but we caution that the quality of the data is variable due to obstructions within the head coil.” Thus, although we found that individuals with noisy retinotopic maps did show a consistent change in gaze position throughout each retinotopic mapping stimuli presentation, we have not found a clear pattern in those with ‘dorsal Y’ maps. That said, as far as we are aware, there is no work suggesting that specific eye movements could lead to localized deviation in retinotopy, i.e., while we find unusual maps in the dorsal portion of the early visual cortex, their respective ventral portion follows the canonical representation (Figure 7, Figure 7—figure supplement 1, and Figure 4—figure supplement 2). All in all, it could be of interest to systematically evaluate gaze position change (and other eye-tracking signal derivatives) and how they relate to pRF modeling accuracy in more detail, which we discuss in our manuscript. Finally, for transparency purposes, we also made an interactive plot of the gaze position data available, as described here https://mne.tools/dev/auto_tutorials/io/70_reading_eyetracking_data.html, in our Jupyter Notebook.

Lines 477-495: “Another potential research direction is determining the extent to which eye movement underlies some of the variability found in retinotopic maps of early visual areas. For example, one could systematically evaluate pRF modeling accuracy as a function of gaze position change (and other eye-tracking signal derivatives). We performed a preliminary analysis of the deviation in gaze position at each time point and averaged across runs of retinotopic mapping stimuli and individuals assigned to each cluster. Supplementary File 6 summarizes the average deviation of gaze position from the fixation point along the X and Y axes for each cluster. In brief, we did not find consistent results across clusters. However, we do not rule out the effect of eye movement on the variability of the retinotopic maps because the eye-tracking data quality is variable and unavailable for some individuals in the HCP retinotopy dataset (Benson et al., 2018), and the clustering quality could be improved (e.g., through manual clustering or the other approaches previously discussed). Another possibility is determining the reliability of these retinotopic maps through connective field modeling (Haak et al., 2013) with unconstrained eye movement data (Tangtartharakul et al., 2023). Yet, we observe that unusual maps in the dorsal portion of the early visual cortex coincide with canonical representations in the ventral portion (Figure 7, Figure 7—figure supplement 1, and Figure 4—figure supplement 2), which is unlikely to be the case for noisy data driven by massive eye movements.”

To get a bit of a better view of the authors' claims I quickly went through the surface maps provided by the NSD paper, since the authors cite this paper as an example study that also showcases the idiosyncrasies reported here. I'm uploading outlines of the most compelling deviations from the canonical V2/V3 organization in this dataset (See https://www.dropbox.com/s/auqjmiz1nghsxv3/Screenshot-Reviewer2.png?dl=0) To summarise; I see clear examples of the patterns outlined by the authors in 4 out of 16 hemispheres (admittedly, slightly conservative an estimate, perhaps). This is below what the authors report. For 2 or 3 of these instances, there is a clear venous eclipse-like effect that could have caused the polar angle patterns to deviate -- we know that this sort of thing is more likely to occur at 7T than at 3T, where retinotopic maps usually look smoother. I don't know what this means in relation to the reported findings, of course, but it could indicate that the more detailed preprocessing and higher single-subject quality of the NSD data decrease the occurrence of the reported patterns.

We thank the reviewer for bringing this to our attention. As per the clustering analysis, we report that 2/3 of the left hemispheres show these unusual maps. However, we did notice it was unclear that we were referring to the left hemispheres in the ‘Abstract,’ so we appropriately addressed that by changing the Abstract. In addition, as highlighted by the reviewer in the uploaded figure, at least 4 out of 8 left hemispheres show unusual maps, which is not at odds with our estimates for the HCP dataset.

Lines 32-36: “Surprisingly, only one-third of individuals had maps that conformed to the expected pattern in the left hemisphere. Visual field sign analysis further revealed that in many individuals the area conventionally identified as dorsal V3 shows a discontinuity in the mirror-image representation of the retina, associated with a Y-shaped lower vertical representation.”

Reviewer #3 (Recommendations for the authors):The work presented here by Ribeiro et al. attempts to quantify the observation that the "standard model" of V1, V2, and V3 matches the average human retinotopic maps but does not consistently match the retinotopic maps of individuals. (In this case, the "standard model" I'm referring to is just the assumption that the boundaries of V1, V2, and V3 topologically resemble those of the wedge-dipole model of Balasubramanian et al., 2002, or the banded double-sech model of Schira et al., 2010). Such non-standard retinotopic organizations in individual subjects are (as the authors note) well documented in other primates, and informal speculation about such non-standard human subjects has historically been common, if inconclusive, among those of us who study human V1-V3. Further, because the visual cortex is one of the few macroscopic brain systems for which neural computations, macroscopic organization, and connectivity are well-modeled, any novel observations about its individual variability are likely to lead to novel hypotheses about brain development and to serve as a template for future analyses. For these reasons, this manuscript is of clear value both to visual neuroscience researchers interested in the details of V1-V3 as well as to neuroscientists in adjacent domains.While the findings of this paper are convincing and leave little doubt that non-standard organizations of V1-V3 not only exist but are common, it falls short of eliminating all possible explanations of such variability. To be clear, this is not primarily a critique of the paper but an observation that the authors' analyses point the field toward many additional questions. While the authors have done an excellent job demonstrating that large-scale deviations in the average curvature and BOLD signal do not explain large-scale variability of the maps, their analyses do not eliminate microscopic features of curvature or vasculature as potential causes. For example, a skeptic might claim that Cluster 2 (Figure 4) can be explained by partial voluming arising from tight curvature in a small patch of the dorsal V2-V3 region or by a blood vessel in the same region. If tight curvature or a vessel were critical to explaining these maps but were confined to a small patch of cortex, the analysis of curvature and BOLD intensity here may have been insufficiently precise to find it. (Of course, given the results of the analysis here, one might be justified in speculating that more detailed examinations of V1-V3 could reveal the "standard" model to be the model that most depend on typical sources of BOLD noise.) It would benefit the paper to include additional discussion of the extent to which the clear differences found in the clustered maps might or might not depend on microscopic features that are beyond the paper's current scope. However, the authors should be lauded for demonstrating conclusively that these differences cannot be easily dismissed.I am particularly intrigued by the clustering analysis performed by the authors, summarized in Figure 4. This figure, along with Supplemental Figure 3, provides compelling evidence that the "Dorsal Y," as the branching lower vertical meridian representation of V1 and V2 is sometimes called, is common and reasonably consistent across individuals. However, I am not entirely convinced that clusters 1, 3, and 5 are capturing substantial deviations in the "standard" model of human retinotopy. It is possible that density-based or hierarchical clustering methods that do not employ any dimension reduction would yield different results (though I am skeptical that the differences would be especially large). Regardless, the fundamental issue with these three clusters is that the topology of the V1-V2-V3 boundaries does not appear terribly different, even when, for example, the intensity of the lower vertical meridian differs. Of course, the intensity of the representation of the meridian may be a feature of the maps worth examining in and of itself, but I suspect that most researchers will be more interested in the clear violation of the topology of visual area boundaries implicit in the "standard" model that one can easily see in Cluster 2. With respect to the differences between Cluster 2 and the other clusters, it seems likely to me that the authors could have captured these differences more easily and accurately by examining the distribution and cortical location of the visual field sign (see for example Tootell et al., 1998, DOI:10.1073/pnas.95.3.811). The most interesting feature of this cluster is arguably that it suggests either (1) that, in these subjects, V2 and V3 are split, with each having a "standard" portion as well as a dorsal-anterior representation; or alternately (2) that at least one (or more likely 2) additional visual areas exist just anterior to dorsal V2 and dorsal to anterior V1. My speculation here – that the patch of cortex bounded by the two branches of the Y (i.e., dorsal to anterior V1 and anterior to dorsal V2) must consist of 2 separate representations – arises from the fact that it contains two separate patches of cortex in which the field signs do not match.

To address some of these concerns, we expanded the discussion about the vasculature effect on retinotopy and the limitations of the LME model to detect the effect of localized (microscopic) features that were not further investigated in our current work.

Lines 453-476: “Retinotopic maps could also vary as a function of data resolution and cortical depth. For example, signal-to-noise ratio (SNR) and partial volume artifacts are directly affected by data resolution (or voxel size); that is, both reduce with voxel size (Hoffmann et al., 2009). While lower SNR might lead to noisier (or less smooth) retinotopic maps (Hoffmann et al., 2009), the gain from the reduced susceptibility to partial volume artifacts will likely result in increased validity of the observed maps. Partial volume artifacts may arise from patches of opposite walls of a sulcus running across a single voxel or even small vessels, leading to inaccurate signals from the combination of different brain regions and tissues. With increasing magnetic field strength, it should be easier to strike the right balance between high SNR and low partial volume artifacts, which is crucial for determining the impact of registration errors due to partial volume artifacts on the variability of retinotopic maps. Moreover, previous studies have also shown that the hemodynamic response function (Puckett et al., 2016) and a spatial pattern of activation (Polimeni et al., 2010) varied across depths in V1. Specifically, Polimeni and colleagues found that a spatial pattern of activation (an ‘M’) becomes clearer from the white matter surface to the mid-thickness surfaces and then deteriorates once again near the pial surface. Altogether, these studies motivate a more thorough investigation of how retinotopy, as measured by fMRI, varies as a function of data resolution and cortical depth and its implication on individual variability in retinotopy. However, it is also important to note that studies of the columnar organization of non-human primates using single-cell recordings have not found any evidence that the receptive field location varies with cortical depth, although the receptive field size changes, being smallest in the middle layers (Hubel and Wiesel, 1974; Rosa et al., 1997).”

Lines 496-548: “Limitations

Although we demonstrate that common modes of deviation from the canonical dorsal V2 and V3 organization exist in the left hemisphere, further analyses are necessary to fully ascertain if such variability is indeed neurogenic or if it could be a result of measurement errors. Amongst potential sources of measurement error is the presence of large veins running adjacent to regions of interest. Accordingly, by using the normalized mean BOLD signal as a proxy for the location of large veins (Boyd Taylor et al., 2019; Kurzawski et al., 2022), studies have shown that voxels near these veins show lower mean BOLD signal, a phenomenon known as the venous eclipse, which affects pRF estimates, for example, in area hV4 (Boyd Taylor et al., 2019; Winawer et al., 2010). As such, deviations in the expected mean BOLD signal could result in deviations from the expected retinotopic organization. Thus, to better understand the potential effects of the presence of large veins on the different levels of variability in visual field representation across early visual areas, we considered both the pair-wise correlations between retinotopic maps and the normalized mean BOLD signal (Figure 3) and the large-scale deviation in the normalized mean BOLD signal as covariates in the LME model (Tables 3 and 4).

In the pair-wise correlation analysis, we found that the polar angle representation is indeed correlated with the normalized mean BOLD signal, and the magnitude of such association varies across visual areas. Specifically, we found a higher correlation (in magnitude) between the polar angle and normalized BOLD signal in ventral visual areas than in dorsal areas. This difference could be explained by ventral areas' proximity to the dural venous sinuses (Winawer et al., 2010), i.e., the transverse sinus, the superior sagittal sinus, and the straight sinus. These venous sinuses are known to introduce artifacts to the BOLD signal, which might lead to changes in retinotopic maps. Importantly, none of these sinuses run near the dorsal V2 and V3 (see Figure 4—figure supplement 2, in which it is possible to observe the likely confluence of sinuses), and both of which show the lowest correlation between polar angle and normalized BOLD signal maps.

Moreover, we modeled large-scale deviation in visual field maps using the large-scale deviation of the normalized mean BOLD signal as a covariate. We did find a significant effect of the normalized mean BOLD signal on the individual variability of eccentricity (Table 4) but not of polar angle maps (Table 3). Thus, to determine the effectiveness of the LME model for uncovering the effects of covariates, we similarly modeled the large-scale variability in polar angle maps of hV4 (not shown in the manuscript). We did not find an effect of the normalized mean BOLD signal on large-scale deviation in polar angle maps in hV4. However, we found a weak correlation between the polar angle and normalized mean BOLD signal (LH: r = -0.06; RH: r = -0.07). This finding does not reconcile with previous reports, i.e., that venous artifact impacts retinotopy. However, a fine-grained analysis of polar angle maps in hV4 indicated the inconsistent effect of the venous artifact on polar angle mapping (Boyd Taylor et al., 2019). In our analysis, though, these results might reflect an inability to appropriately parcellate hV4 at the individual level when using an atlas-based parcellation; while early visual areas are more consistently found at specific spatial locations, this is not the case for other visual areas of which spatial location and extent seem to vary across participants. Therefore, although we did not find a significant effect of the normalized mean BOLD signal in our LME model, it does not mean that the macro- and microvasculature do not affect retinotopy. For the former, future research might consider a fine-grained analysis of topographic deviations, such as the one reported by Boyd Taylor et al. (2019). For the latter, given that the mean BOLD signal is only used as a proxy for the location of large veins, a more detailed analysis of the microvasculature might require other imaging data, such as high-resolution time-of-flight magnetic resonance angiography data (Bollmann et al., 2022).”

Another possibility that I would have liked to see discussed is the likelihood that the "Dorsal Y" organization of Cluster 2 appears in many (more) subjects but does not appear at a consistent location with respect to eccentricity. To this end, I believe that Supplemental Figure 2 would have been more interesting had it been calculated not as a new set of clusters based on eccentricity but instead as the average eccentricity map of each of the clusters from Figure 4 that were based on the polar angle. My own observation (which I would like to emphasize is purely anecdotal) is that the eccentricity maps of subjects whose polar angle maps resemble those of Cluster 2 tend to bend sharply near the junction of the "y". I find myself wondering if the subjects in Cluster 2 have a similar feature in their eccentricity maps, whether the junction of the "y" of Cluster 2 tends to occur near the same eccentricity across subjects, and whether the cortical magnification functions of such subjects differ substantially from those who do not have this feature. The size of the PRFs in these regions might also be illuminating---do they scale with eccentricity in a manner that matches the rest of dorsal V2 and V3?

We included a figure supplement (Figure 4—figure supplement 2) with a set of average maps of each cluster from Figure 4c, i.e., the cluster assignment was based on the clustering analysis with polar angle maps from the dorsal portion of the early visual cortex and left hemisphere. Based on the new figure, we could not find evidence that the polar angle mapping variability coincides with variability in eccentricity.

Overall, despite (and arguably because of) the many questions that this paper exposes, it is of substantial value to the community. Its results are straightforward and well-supported, and it provides a valuable starting point from which further refinements of the organization of the visual cortex can be studied.Overall this paper is compelling and does not require substantial revision or refinement in order to be of great value to the field. To be frank, I have been hoping for several years that someone would publish a clear analysis of the dorsal retinotopic maps in V1-V3, and am pleased to see a conclusive demonstration of their variability finally appear. In my public review, I have pointed out many questions and implications that arise from these findings, and I believe that mentioning some of these implications would improve the paper. However, I do not believe that answering them is necessary for the paper – merely that discussion may be helpful for researchers who do not work on these maps closely on a regular basis.

We thank the reviewer for the interesting remarks made in the public review, and we have included responses to some of those above.

One concrete suggestion I do have for the authors is that a cursory examination of the field sign may lead to clearer clusters and thus a clearer narrative of the data in Figure 4. That said, the field sign can be tricky – such an analysis would likely rest at least somewhat on light smoothing of the PRF parameters and may become difficult in ways I have not anticipated. Similarly, a brief analysis of the patch of cortex bounded by the arms of the "dorsal Y" in Cluster 2 would be of substantial interest and may help determine what kind of deviation from the traditional maps this cluster represents.

We thank the reviewer for these suggestions. In the manuscript, we expanded our analysis of the “dorsal Y” by including the representation of polar angle maps’ gradients and their corresponding visual field sign representations, aiming to elucidate the kind of deviation from the traditional maps this cluster represents. In brief, the visual field sign analysis suggests that the area identified as dorsal V3 shows a discontinuity in the canonical mirror-image representation in individuals where the canonical type of representation is not evident.

Lines 800-821: “Visual field sign analysis

Lastly, we further examined unusual retinotopic maps to elucidate the kind of deviation from the canonical maps they represent. We performed a visual field sign analysis (Sereno et al., 1995, 1994), which combines both polar angle and eccentricity maps into a unique representation of the visual field as either a non-mirror-image (like V2) or a mirror-image representation of the retina (like V1). Since the left hemisphere represents the right visual field, which in polar angle includes 0⁰-90⁰ (upper right visual field) and 270⁰- 360⁰ (lower right visual field), we shifted the polar angle values so the point of wrap-around (from 360⁰ to 0⁰) was positioned at the horizontal meridian in the contralateral hemifield, avoiding the discontinuous representation between 360⁰ and 0⁰. Then, we interpolated the sparse and flattened polar angle and eccentricity maps onto a regular x-y grid using SciPy (Virtanen et al., 2020). Next, we determined the gradient of polar angle and eccentricity maps, mathematically expressed as:

Δfθ=∂fθ∂xi^+∂fθ∂yj^ and Δfr=∂fr∂xi^+∂fr∂yj^

where fθ is the polar angle map and fr the eccentricity, using the NumPy gradient numerical method (Harris et al., 2020). Finally, the angle between the polar angle and eccentricity maps’ gradient vectors was determined at each x-y coordinate. If the angle between the gradient vectors is between 0 and π, by convention, the cortical patch is a mirror-image representation of the retina; otherwise, it is a non-mirror-image. After binarizing the angle projection, we can conveniently infer borders between visual areas because adjacent areas often have the opposite visual field sign (Sereno et al., 1995; but see Yu et al. 2020 for caveat).”

Lines 291-340: Visual field sign analysis for delineating visual area boundaries

“Finally, we further examined retinotopic maps with a Y-shaped lower vertical representation, i.e., those primarily assigned to cluster 2, to elucidate the kind of deviation from the canonical maps they represent (Figure 7 and Figure 7—figure supplement 1). To do so, we performed a visual field sign analysis (Sereno et al., 1995, 1994), which combines both polar angle and eccentricity maps into a unique representation of the visual field as either a non-mirror-image or a mirror-image representation of the retina (Figure 7a; see Materials and methods). With such representation, we can directly infer visual area parcellation.

Figure 7b shows polar angle gradients in a ‘streamline’ representation and their respective visual field sign representation for two participants with canonical and four with Y-shaped lower vertical representations. While visual area boundaries in early visual cortex are conventionally identified by reversals in the progression of the polar angle values – or changes in the direction of polar angle gradients, it is unclear how to delineate boundaries in the dorsal portion of polar angle maps in those participants with non-canonical maps (note that their respective ventral portion followed the classical representation), but not on those with canonical maps. However, with the visual field sign representation, the boundaries delineating dorsal V2 in those participants with non-canonical maps are more explicit, and it reveals that the area identified as dorsal V3 shows a discontinuity in the expected mirror-image representation. Such representation has been proposed as the ‘incomplete-V3’ model of the third-tier cortex for the macaque (Angelucci and Rosa, 2015) and other similar models for the owl monkey (Sereno et al., 2015) and the marmoset monkey (Rosa and Tweedale, 2000). Figure 7—figure supplement 1 shows five more examples of polar angle maps with unusual Y-shaped lower vertical representations and five other examples with a truncated V3 boundary. While individuals with unusual Y-shaped lower vertical representations have a discontinuity in the canonical mirror-image representation of the retina in dorsal V3, individuals with a truncated V3 do not show such a discontinuity. Altogether, our findings may suggest that, at least in humans, the canonical model does not oppose other models established for non-human primates; these models coexist and reflect common modes of variability in the retinotopic mapping of early visual areas.”

I would also suggest that the authors include some discussion of how the human clusters found in this analysis compare to the 6 clusters found in non-human primates. I don't think that much ink needs to be spent on this, but it struck me as odd that the non-human primate clusters would be cited as motivation for the clustering parameters in humans and then never discussed more carefully.

As indicated by the reviewer, we selected this number of clusters as there are at least five different models of third-tier visual cortex organization in non-human primates. However, note that these models of the third-tier cortex are a guess at the size of the possibility space and do not reflect known inter-individual differences in non-human primates. Specifically, the non-human primate data is described by competing theories for how the third-tier visual cortex is organized, and each explains different aspects of parts of data sets that are far less complete than the HCP data. Besides, whether the areal boundaries in this region show significant individual variability has yet to be studied systematically in non-human primates (lines 583-588). We clarified this aspect in our methods.

Lines 795-797: ***“***Note, however, that this selection is simply a speculation of the possibility space and do not reflect known inter-individual differences in non-human primates.”